# *In vitro* interactions of berbamine hydrochloride and azoles against *Aspergillus fumigatus*

Heng Zhang,[1] Yizheng Zhou,[2] Hanmo Yang,[3] Xinyi Tao,[3] Yinping Chen,[1] Fuqiang Dong,[4] Yi Sun[1]

**ABSTRACT** The growing resistance of *Aspergillus* to azoles poses a significant challenge in treating invasive fungal infections. This study aimed to investigate the synergistic effect of berbamine hydrochloride (BBM) combined with azoles in treating *Aspergillus fumigatus* and explore the role of efflux pump inhibition in this synergy. The efficacy of combining BBM with itraconazole (ITC), voriconazole (VOR), and posaconazole (POS) was tested against 69 *A. fumigatus* strains that had been identified using the M38-A3 broth microdilution method. quantitative reverse transcription PCR (RT-qPCR) was used to measure gene expression related to synergy, while flow cytometry was employed to assess mitochondrial reactive oxygen species (ROS) levels, and Rhodamine 6G exocytosis assays were performed to quantify efflux pump activity. BBM alone showed no significant antifungal activity. BBM combined with POS exhibited synergy against 66 strains (95.7%), while two clinical isolates (Af05/Af08) and one defective strain (Δ*cdr1B*) showed no synergy. Synergy with ITC was observed in three strains (4.3%), but not with VOR. In the non-synergistic Af05 and Af08 strains, the expression of the *cdr1B* gene was significantly lower compared to wild-type (WT) strains. ROS levels increased significantly in WT with POS and BBM combination therapy, but not in the defective strains. Glucose uptake was also reduced in the POS-BBM combination. BBM enhances azole sensitivity in *A. fumigatus* primarily by inhibiting the *cdr1B*-mediated efflux pump, supported by reduced Rhodamine 6G exocytosis and synergy loss in *cdr1B*-deficient strains. ROS accumulation and metabolic disruption may further contribute to this synergy. Targeting efflux pumps with BBM provides a novel strategy to combat azole resistance.

**IMPORTANCE** The combination of berbamine hydrochloride and posaconazole effectively enhances azole sensitivity in *Aspergillus fumigatus* by reducing efflux pump activity and increasing reactive oxygen species levels. The findings offer a promising strategy to combat azole resistance in invasive fungal infections.

**KEYWORDS** berbamine hydrochloride, *Aspergillus fumigatus*, posaconazole, synergistic

Aspergillus fumigatus represents a significant opportunistic pathogen in humans and is the primary cause of invasive aspergillosis (IA) (1). This fungus has the capacity to infect a multitude of human organs and tissues, including the lungs, sinuses, skin, soft tissues, bones, and the cranium, in addition to various systemic tissues (2). Severe infections can pose a significant risk to patient survival. Annually, there are approximately 250,000 cases of IA and 3 million cases of chronic pulmonary aspergillosis globally (3). In individuals undergoing immunosuppressive therapy or those with compromised immune systems, the mortality rate from invasive pulmonary aspergillosis can reach as high as 90% (4). At present, the principal treatment for infections caused by *A. fumigatus* is based on azole drugs (3). However, the rapidity of environmental change and the growth of high-risk populations have resulted in the development of resistance to all approved systemic antifungal azoles (4). Nevertheless, the prolonged utilization

**Peer Reviewer** Payal Gupta, Graphic Era Deemed to be University, Dehradun, Uttarakhand, India

Address correspondence to Yi Sun, jzzxyysy@163.com.

Heng Zhang and Yizheng Zhou contributed equally to this article. The order of names was determined based on Heng Zhang's primary responsibility in manuscript writing and experimental design optimization.

The authors declare no conflict of interest.

See the funding table on p. 11.

of azole pharmaceuticals has resulted in an elevated prevalence of resistance in *A. fumigatus*, particularly among strains that are resistant to pantoprazole. The efficacy of voriconazole (VOR) is only 52.8%, while amphotericin B, a conventional antifungal agent, has an efficacy of just 31.6% (5, 6). In addition, itraconazole (ITC) has an efficacy of 43%–76%, and posaconazole (POS) has an efficacy of 44%–61% (7). It is worth noting that the treatment rate for invasive fungal infections in hematopoietic stem cell transplant patients with POS is 57.8% (8). Treatment failure is frequently attributable to the host's intolerance or resistance to antifungal drugs (6). One of the causes of this resistance is the alteration in efflux pump activity. These pumps are involved in the active extrusion of antimicrobial molecules, including azoles. Multidrug resistance (MDR) efflux transporter genes from both the ATP-binding cassette (ABC) and major facilitator superfamily (MFS) classes have been shown to play a clinically significant role in various pathogenic fungi (9, 10). Sequence analysis suggests that *A. fumigatus* contains 278 different MFS and 49 ABC transporters (11). Several studies have described the role of *A. fumigatus* MDR pumps, showing their association with increased resistance to ITC (12, 13). In conclusion, these MDR efflux pumps influence the sensitivity of the fungus to azoles.

In such a scenario, combination therapy, which has the potential to enhance the efficacy of currently applied antifungals and reduce the likelihood of the development of resistance, represents a valuable and promising alternative to drug monotherapy.

Berbamine hydrochloride (BBM; Fig. S1), a natural compound derived from Berberis, is a bis-benzylisoquinoline alkaloid that has been extensively utilized in China for the treatment of leukopenia over the past several decades (14, 15).

BBM has been demonstrated to exhibit a range of bioactivities, including anti-inflammatory, anti-hypertensive, and anti-arrhythmic effects (16–18). Recently, a number of studies have demonstrated the anti-tumor effects of BBM in a number of different cancers, including myeloma, prostate cancer, lung cancer, pancreatic cancer, liver cancer, and gastric cancer (19–21). However, there have been no reports on the combination of BBM with other drugs. Furthermore, studies have demonstrated that BBM can effectively induce intracellular reactive oxygen species (ROS) levels (22–26). Antifungal drugs, such as miconazole (azole), caspofungin, and amphotericin B (polyene), primarily function by inducing the production of ROS in fungal (biofilm) cells (27). ROS is an essential component of the host defence against fungal infections (28). The aim of this study was to investigate the synergistic effect of BBM combined with azole drugs against *A. fumigatus in vitro*, providing a potential high-efficiency strategy for combating IA.

## MATERIALS AND METHODS

### Fungal strains

A total of 69 strains of *A. fumigatus* were tested, including the wild-type (WT) strain, 37 clinical isolates, and 31 genetically deficient strains (Table S1). The genetically deficient strains can be divided into those missing MFS transporters (20), ABC transporters (5), and genes related to the Tricarboxylic Acid (TCA) cycle (6). *Candida parapsilosis* (ATCC 22019) and *Aspergillus flavus* (ATCC 204304) were included to ensure quality control. The construction of the deficient strains was achieved using high-throughput gene knockout based on the principle of homologous recombination (29). Details on the identification of clinical isolates and the construction of genetically deficient strains are provided in the supplementary file.

### Antifungals and chemical agents

All tested agents, including BBM, ITC, VOR, and POS, were purchased in powder form from Aladdin Biochemical Technology Co., Ltd., Shanghai, China, and diluted in dimethyl sulfoxide (≥99.9%) as stock solutions (6,400 µg/mL).

## Testing the *in vitro* synergy of BBM and azoles

Susceptibility testing was performed according to the broth microdilution chequerboard procedure based on the Clinical and Laboratory Standards Institute (CLSI) M27-A3 (30) standard and previously published protocols (31, 32). Conidia harvested from cultures grown for 3 days on Sabouraud Dextrose Agar (SAB) solid medium were suspended in sterile distilled water containing 0.03% Triton and diluted to a concentration of 2–5 $\times 10^6$ cfu/mL, which were then diluted 100 times in RPMI 1640 to achieve a twofold suspension more concentrated than the density needed or to approximately 1–3 $\times 10^4$ cfu/mL (32).

The working concentration ranges of BBM, ITR, VRC, and POS were 0.5–32 µg/mL, 0.06–8 µg/mL, 0.06–8 µg/mL, and 0.03–4 µg/mL, respectively. A 50 µL of BBM with serial dilutions was inoculated in the horizontal direction, and another 50 µL of azoles with serial dilutions was inoculated in the vertical direction on the 96-well plate, which contained 100 µL prepared inoculum suspension. Interpretation of results was performed after incubation at 35°C for 48 h. The MICs applied for the evaluation of effects against *A. fumigatus* were determined as the lowest concentration resulting in 100% inhibition of growth (32). The fractional inhibitory concentration index (FICI) is calculated by the formula: FICI = (Ac/Aa) + (Bc/Ba), where Ac and Bc are the MICs of antifungal drugs in combination, and Aa and Ba are the MICs of antifungal drugs A and B alone (33). A FICI of ≤0.5 is classified as synergy, a FICI of >0.5 to ≤4 indicates no interaction (indifference), and a FICI of >4 indicates antagonism (34). All tests were performed in triplicate.

## Detection of ROS content

*A. fumigatus* conidia were collected after 48 hours of incubation on SAB solid medium at 35°C and resuspended in 1640 liquid medium to a final concentration of 5 $\times 10^4$ cfu/mL. All samples were incubated at 37°C in a shaking incubator at 130 rpm for 16 hours. During the incubation period, drug treatments were applied as needed. At the end of the incubation, 2,7-dichlorodihydrofluorescein diacetate (Yeasen Biotechnology, Shanghai, China) was added to each sample and incubated at 37°C for 1 hour to detect intracellular ROS levels. Flow cytometry data acquisition was performed using a Beckman Cytomics FC 500 BD FACSCanto II, and data analysis was conducted using FlowJo v10 software. The excitation wavelength was set at 488 nm, and the emission wavelength was 525 nm.

## RT-qPCR detection of relevant gene expression

Total RNA was extracted using TRleasy (Yeasen Biotechnology, Shanghai, China) and reverse transcribed into cDNA using Hifair III 1st Strand cDNA Synthesis SuperMix (Yeasen Biotechnology, Shanghai, China) for each group, following the manufacturer's recommendations. The expression of the *cdr1B* gene under different treatments was then detected (Table S3). The relative gene expression level ($2^{-\Delta\Delta CT}$) was calculated using the actin housekeeping gene control (35).

## Rhodamine 6G efflux by *A. fumigatus* cells

The efflux of rhodamine 6G (R6G) from intact *A. fumigatus* cells was determined by adapting the method described by Kolaczkowski et al. (36) for each group. Conidia cells from SAB cultures in the exponential growth phase (OD600, 0.5) were collected by centrifugation (3,000 g, 5 min, and 20°C) and washed three times with water. The cells were resuspended at a concentration of 0.5 $\times 10^6$ to 1.0 $\times 10^7$ cells/mL in HEPES-NaOH (50 mM; pH 7.0) containing 5 mM 2-deoxyglucose and 10 µM R6G. Cell suspensions were incubated at 30°C with shaking (200 rpm) for 90 min to allow rhodamine accumulation under glucose starvation conditions. The starved cells were washed twice in HEPES-NaOH, and portions (400 µL) were incubated at 30°C for 5 min before the addition of glucose (2 mM) to initiate rhodamine efflux. At specified intervals after the addition of glucose, the cells were removed by centrifugation, and triplicate 100 µL volumes of the

cell supernatants were transferred to the wells of 96-well flat-bottom microtiter plates (BKMAM Biotechnology, Hunan, China). The rhodamine fluorescence of the sample was measured by a microplate reader (ALLSHENG, Wuhan, China). The microplate reader was read at the excitation wavelength of 529 nm.

## Data processing software

The GraphPad Prism 8.0 was used for statistical analysis. The results were presented as mean ± SD or mean ± SEM. Student $t$‑test, one‑way Analysis of Variance (ANOVA), or two‑way ANOVA was employed for statistical comparison. Statistical significance was determined with 95% (*, $P < 0.05$), 99% (**, $P < 0.01$), 99.9% (***, $P < 0.001$), and 99.99% (****, $P < 0.0001$) confidence intervals.

## RESULTS

### Identification results of isolated strains

The identification results showed that all 34 clinical isolates were *A. fumigatus*. All the sequences of the analyzed strains were uploaded to GenBank (PP069948-PP070390).

### *In vitro* interactions between BBM and azoles against *A. fumigatus*

The MIC ranges of individual tested agents against planktonic *A. fumigatus* isolates were >32 µg/mL for BBM, 0.125–4 µg/mL for ITC, 0.125–4 µg/mL for VRC, and 0.125–2 µg/mL for POS. BBM alone did not show anti-*A. fumigatus* activity. Specifically, in clinical isolates ($n = 37$), MIC ranges were 0.125–4 µg/mL for ITC, 0.25–1 µg/mL for VRC, and 0.125–2 µg/mL for POS. Among the defective strains in the MFS transporter ($n = 20$), MIC ranges were 0.125–0.5 µg/mL for ITC, 0.125–1 µg/mL for VRC, and 0.125–1 µg/mL for POS. Among the defective strains in the ABC transporter ($n = 5$), MIC ranges were 0.25–0.5 µg/mL for ITC, 0.125–1 µg/mL for VRC, and 0.125–0.5 µg/mL for POS. Among Tricarboxylic Acid Cycle (TCA) cycle-related gene-deficient strains ($n = 6$), MIC ranges were 0.125–2 µg/mL for ITC, 0.25–4 µg/mL for VRC, and 0.25–1 µg/mL for POS (Table 1). When BBM was used in combination with ITR, VOR, and POS, it synergized mainly with POS: high rates of synergistic effects were shown WT, 35 clinical isolates (94.4%), and 30 genetically defective strains (96.8%; total synergistic rate 95.7%). The effective MIC ranges of BBM and ITC in the BBM-ITC combination were 8–64 µg/mL and 0.0625–

**TABLE 1** MIC values distribution of tested strains

| Azole | MIC | Non-knockout strains | Knockout strains | Total |
|---|---|---|---|---|
| ITC | 0.125 | 2 | 4 | 6 |
| | 0.25 | 17 | 12 | 29 |
| | 0.5 | 6 | 13 | 19 |
| | 1 | 7 | 1 | 8 |
| | 2 | 3 | 1 | 4 |
| | 4 | 3 | 0 | 3 |
| VOR | 0.125 | 0 | 2 | 2 |
| | 0.25 | 12 | 4 | 16 |
| | 0.5 | 21 | 11 | 32 |
| | 1 | 5 | 12 | 17 |
| | 2 | 0 | 1 | 1 |
| POS | 4 | 0 | 1 | 1 |
| | 0.125 | 2 | 6 | 8 |
| | 0.25 | 17 | 10 | 27 |
| | 0.5 | 8 | 13 | 21 |
| | 1 | 7 | 2 | 9 |
| | 2 | 4 | 0 | 4 |
| | 4 | 0 | 0 | 0 |

2 µg/mL, respectively. In the BBM combination, the effective working ranges of BBM and VRC were 16–64 µg/mL and 0.0625–4 µg/mL, respectively, while in the BBM-POS combination, the effective MIC ranges of BBM and POS were 4–16 µg/mL and 0.03125–0.25 µg/mL, respectively. Of the 31 genetically gene-deficient strains, Δ*cdr1B* produced a significant reversal of the synergistic effect on POS synthetic BBM. In addition, no antagonistic effects were detected in all combinations. Notably, some of the synergistic effects of diminished sensitivity to azoles were also detected in the wild-type strains (Af19, Af23, and Af28-36) and knockout strains (Δ*mfs05* and Δ*tca1*; Table 2).

## Expression of *cdr1B* gene

As BBM did not exhibit significant synergy with ITC or VOR, but showed strong synergy with POS, and since the synergy observed in the BBM-POS combination was lost in the Δ*cdr1B*, *cdr1B* is likely a key target in the synergistic mechanism of BBM and POS. From the 36 clinical isolates, Af05 and Af08 did not produce a synergistic effect, possibly due to inherently low expression levels in these strains. Therefore, the expression levels of the *cdr1B* in the clinical isolates Af05 and Af08 were also examined. The results showed that compared to WT and isolated strain Af293, the initial expression levels of the *cdr1B* in Af05 and Af08 were significantly reduced. Furthermore, under the combined stimulation of BBM and POS, the expression levels of *cdr1B* in WT and Af293 were significantly decreased, but no changes were detected in Af05 and Af08, suggesting that low expression or non-expression of *cdr1B* may be the reason for reversing the synergistic effect (Fig. 1).

## Analysis of Cdr1B information

The *cdr1B* (AFUA_1G14330) belongs to the ABC transporter superfamily, specifically the ABCG family. Gene ID: 3509814, composed of 4,760 base pairs, encoding 630 amino acids, located on chromosome 1. It has one structural domain. The *cdr1B* gene enables the extrusion of azole-class drugs from cells by utilizing ATP: ATP + Zoles(in) + $H_2O$ = ADP + Zoles(out) + $H^+$ + phosphate (Fig. 2).

## ROS levels after drug stimulation

Only the BBM-POS combination demonstrated a significant synergistic effect. Therefore, we further investigated the impact of the BBM-POS combination on ROS levels. In the absence of POS or BBM, there was no significant difference in the levels of reactive ROS among the three fungal strains (WT, Af293, and Af05), except for strain Af08 and Δ*cdr1B*, which exhibited relatively lower ROS levels. Upon the addition of POS, three strains (WT, Af293, and Af05) exhibited an increase in ROS levels, while two strains (Af08 and Δ*cdr1B*) showed no change in ROS content. Subsequently, under the combined stimulation of BBM and POS, a notable increase in ROS levels was detected in WT and Af293. However, in the three strains that did not produce a synergistic effect, there was no further increase in ROS levels; in fact, a decrease in ROS levels was observed in Af05. In summary, in the three strains capable of reversing the synergistic effect, the combined stimulation of BBM and POS did not further enhance ROS production. Conversely, in WT and Af293, which exhibited a synergistic effect, there was a further elevation in ROS levels (Fig. 3).

## POS combined with BBM reduces the efflux pump

Given that the target of BBM combined with POS is *cdr1B*, Cdr1B is a representative efflux pump protein, indicating that the synergistic effect of BBM in combination with POS is due to the alteration of efflux pump activity. Consequently, the present study proceeded to examine the alteration of efflux pump activity. As expected, the addition of BBM leads to a significant reduction in the external displacement of R6G in WT and Af293. Specifically, at 9, 12, and 15 minutes after treatment, the fluorescence intensity of WT cells decreased by 13, 37, and 40 times, while that of Af293 cells decreased by 10, 28,, and 121 times, respectively. These findings indicate that the combination of BBM and POS

**TABLE 2** MIC corresponding to the combined use of BBM and azoles against *A. fumigatus*[c]

| Species | MIC (µg/mL)[a] | | | | | | |
|---|---|---|---|---|---|---|---|
| | Agent alone | | | | Combination[b] | | |
| | BBM | ITC | VOR | POS | BBM/ITC | BBM/VOR | BBM/POS |
| WT | >32 | 0.25 | 0.5 | 0.25 | 16/0.25 (1.25, I) | 32/0.5 (1.5, I) | 8/0.0625 (0.375, S) |
| Af293 | >32 | 0.25 | 0.5 | 0.5 | 16/0.5 (2.25, I) | 16/0.25 (0.75, I) | 4/0.125 (0.3125, S) |
| Af01 | >32 | 0.25 | 0.5 | 0.25 | 32/0.125 (1, I) | 32/0.0625 (0.625, I) | 8/0.0625 (0.375, S) |
| Af02 | >32 | 0.25 | 0.5 | 0.125 | 16/0.0625 (0.5, S) | 32/0.0625 (0.625, I) | 8/0.03125 (0.375, S) |
| Af03 | >32 | 0.125 | 1 | 0.25 | 16/0.25 (2.25, I) | 32/0.0625 (0.5625, I) | 8/0.0625 (0.375, S) |
| Af04 | >32 | 0.25 | 0.5 | 0.25 | 32/0.5 (2.5, I) | 32/0.25 (1, I) | 4/0.0625 (0.3125, S) |
| Af05 | >32 | 0.25 | 0.5 | 0.25 | 32/0.25 (1.5, I) | 16/0.5 (1.25, I) | 8/0.125 (0.625, I) |
| Af06 | >32 | 0.25 | 0.5 | 0.25 | 32/0.125 (1, I) | 32/0.0625 (0.625, I) | 8/0.0625 (0.375, S) |
| Af07 | >32 | 0.25 | 0.5 | 0.125 | 8/0.0625 (0.375, S) | 32/0.0625 (0.625, I) | 4/0.03125 (0.3125, S) |
| Af08 | >32 | 0.25 | 0.5 | 0.25 | 16/0.25 (1.25, I) | 16/0.25 (0.75, I) | 8/0.25 (1.125, I) |
| Af09 | >32 | 0.125 | 1 | 0.25 | 32/0.125 (1.5, I) | 16/0.25 (1.5, I) | 4/0.0625 (0.3125, S) |
| Af10 | >32 | 0.25 | 0.5 | 0.25 | 16/0.5 (2.25, I) | 32/0.0625 (0.625, I) | 4/0.0625 (0.3125, S) |
| Af11 | >32 | 0.25 | 1 | 0.5 | 32/0.0625 (0.75, I) | 32/0.25 (0.75, I) | 4/0.125 (0.3125, S) |
| Af12 | >32 | 0.25 | 0.5 | 0.25 | 32/0.125 (1, I) | 32/0.125 (0.75, I) | 8/0.0625 (0.3125, S) |
| Af13 | >32 | 0.25 | 0.5 | 0.25 | 32/0.25 (1.5, I) | 32/0.25 (1.5, I) | 4/0.0625 (0.3125, S) |
| Af14 | >32 | 0.25 | 0.5 | 0.5 | 32/0.125 (1, I) | 32/0.125 (1, I) | 4/0.125 (0.3125, S) |
| Af15 | >32 | 0.5 | 0.5 | 0.25 | 32/0.125 (0.75, I) | 32/0.25 (0.75, I) | 8/0.0625 (0.375, S) |
| Af16 | >32 | 0.25 | 0.5 | 0.5 | 32/0.25 (1.75, I) | 32/0.25 (1, I) | 8/0.125 (0.375, S) |
| Af17 | >32 | 0.5 | 0.25 | 0.25 | 32/0.125 (0.5, I) | 32/0.125 (1, I) | 8/0.03125 (0.25, S) |
| Af18 | >32 | 1 | 0.25 | 0.5 | 32/1 (1.5, I) | 8/0.125 (0.625, I) | 4/0.125 (0.3125, S) |
| Af19 | >32 | 0.25 | 0.25 | 1 | 32/0.25 (1.5, I) | 32/0.25 (1.5, I) | 8/0.125 (0.25, S) |
| Af20 | >32 | 0.5 | 0.25 | 0.25 | 32/0.25 (1, I) | 8/0.125 (0.625, I) | 16/0.0625 (0.5, S) |
| Af21 | >32 | 1 | 0.5 | 0.25 | 16/0.5 (0.75, I) | 32/0.25 (1, I) | 4/0.0625 (0.3125, S) |
| Af22 | >32 | 0.5 | 0.25 | 0.5 | 32/0.25 (1, I) | 8/0.25 (1.125, I) | 8/0.125 (0.375, S) |
| Af23 | >32 | 0.25 | 0.5 | 1 | 32/0.25 (1.5, I) | 32/0.25 (1, I) | 4/0.0625 (0.125, S) |
| Af24 | >32 | 1 | 0.25 | 0.25 | 32/0.25 (0.75, I) | 8/0.125 (0.625, I) | 4/0.0625 (0.3125, S) |
| Af25 | >32 | 0.5 | 0.25 | 0.25 | 16/0.5 (1.25, I) | 32/0.25 (1.5, I) | 4/0.0625 (0.3125, S) |
| Af26 | >32 | 1 | 0.5 | 0.5 | 32/0.5 (1, I) | 16/0.25 (0.75, I) | 4/0.25 (0.5625, S) |
| Af27 | >32 | 0.5 | 0.25 | 0.5 | 16/0.25 (0.75, I) | 8/0.25 (1.125, I) | 8/0.125 (0.375, S) |
| Af28 | >32 | 2 | 0.25 | 1 | 32/1 (1, I) | 32/0.25 (1.5, I) | 8/0.25 (0.375, S) |
| Af29 | >32 | 1 | 0.25 | 2 | 32/0.25 (0.75, I) | 32/0.25 (1.5, I) | 8/0.5 (0.375, S) |
| Af30 | >32 | 4 | 1 | 2 | 8/1 (0.375, S) | 1/0.5 (0.515625, I) | 16/0.25 (0.375, S) |
| Af31 | >32 | 2 | 0.5 | 1 | 32/0.5 (0.75, I) | 1/0.25 (0.515625, I) | 8/0.25 (0.375, S) |
| Af32 | >32 | 1 | 0.5 | 1 | 32/0.25 (0.75, I) | 1/1 (2.015625, I) | 16/0.25 (0.5, S) |
| Af33 | >32 | 4 | 0.5 | 2 | >32/2 (1.5, I) | 4/1 (2.0625, I) | 16/0.5 (0.5, S) |
| Af34 | >32 | 2 | 0.25 | 1 | 16/1 (0.75, I) | 32/0.25 (1.5, I) | 8/0.125 (0.25, S) |
| Af35 | >32 | 1 | 0.25 | 1 | 32/0.25 (0.75, I) | 4/0.5 (2.0625, I) | 8/0.25 (0.375, S) |
| Af36 | >32 | 4 | 1 | 2 | 16/2 (0.75, I) | 32/1 (1.5, I) | 16/0.5 (0.5, S) |
| △mfs01 | >32 | 0.25 | 0.125 | 0.25 | 32/0.25 (1.5, I) | 32/0.0625 (1, I) | 4/0.0625 (0.3125, S) |
| △mfs02 | >32 | 0.25 | 0.5 | 0.25 | 32/0.125 (1, I) | 32/0.125 (0.75, I) | 8/0.0625 (0.375, S) |
| △mfs03 | >32 | 0.25 | 0.5 | 0.5 | 16/0.125 (0.75, I) | >32/0.25 (1.5, I) | 8/0.0625 (0.25, S) |
| △mfs04 | >32 | 0.125 | 0.5 | 0.5 | 16/0.125 (1.25, I) | 32/0.125 (0.75, I) | 4/0.125 (0.3125, S) |
| △mfs05 | >32 | 0.25 | 1 | 1 | 32/0.25 (1.5, I) | >32/0.125 (1.125, I) | 4/0.125 (0.1875, S) |
| △mfs06 | >32 | 0.5 | 0.25 | 0.25 | >32/0.125 (1.25, I) | >32/0.25 (2, I) | 8/0.0625 (0.375, S) |
| △mfs07 | >32 | 0.125 | 1 | 0.5 | 16/0.0625 (0.75, I) | 32/0.125 (0.625, I) | 8/0.125 (0.375, S) |
| △mfs08 | >32 | 0.25 | 1 | 0.125 | >32/0.25 (2, I) | 32/0.125 (0.625, I) | 8/0.03125 (0.375, S) |
| △mfs09 | >32 | 0.25 | 0.5 | 0.5 | 32/0.125 (1, I) | >32/0.25 (1.5, I) | 4/0.0625 (0.1875, S) |
| △mfs10 | >32 | 0.5 | 1 | 0.5 | >32/0.0625 (1.125, I) | 32/0.125 (0.625, I) | 8/0.125 (0.375, S) |
| △mfs11 | >32 | 0.5 | 1 | 0.5 | 32/0.0625 (0.625, I) | 32/0.125 (0.625, I) | 8/0.0625 (0.25, S) |
| △mfs12 | >32 | 0.25 | 0.5 | 0.125 | 16/0.125 (0.75, I) | 32/0.25 (1, I) | 8/0.03125 (0.375, S) |

TABLE 2 MIC corresponding to the combined use of BBM and azoles against *A. fumigatus*[c] (*Continued*)

| Species | MIC (µg/mL)[a] | | | | | | |
|---------|------|------|------|------|------|------|------|
| | **Agent alone** | | | | **Combination**[b] | | |
| | **BBM** | **ITC** | **VOR** | **POS** | **BBM/ITC** | **BBM/VOR** | **BBM/POS** |
| △*mfs13* | >32 | 0.5 | 1 | 0.5 | >32/0.0625 (1.125, I) | >32/0.125 (1.125, I) | 8/0.125 (0.375, S) |
| △*mfs14* | >32 | 0.25 | 1 | 0.5 | 16/0.25 (1.25, I) | 32/0.125 (0.625, I) | 16/0.0625 (0.375, S) |
| △*mfs15* | >32 | 0.5 | 0.5 | 0.25 | 32/0.0625 (0.625, I) | 32/0.25 (1, I) | 8/0.03125 (0.25, S) |
| △*mfs16* | >32 | 0.5 | 1 | 0.125 | >32/0.125 (1.25, I) | 32/0.125 (0.625, I) | 8/0.03125 (0.375, S) |
| △*mfs17* | >32 | 0.25 | 0.5 | 0.25 | 32/0.0625 (0.75, I) | 32/0.125 (0.75, I) | 8/0.0625 (0.375, S) |
| △*mfs18* | >32 | 0.125 | 1 | 0.25 | >32/0.125 (2, I) | 16/0.5 (0.75, I) | 4/0.03125 (0.1875, S) |
| △*mfs19* | >32 | 0.5 | 0.5 | 0.25 | 32/0.0625 (0.625, I) | >32/0.125 (1.25, I) | 8/0.0625 (0.375, S) |
| △*mfs20* | >32 | 0.25 | 1 | 0.125 | 32/0.0625 (0.75, I ) | 32/0.125 (0.625, I) | 8/0.03125 (0.375, S) |
| △*abc1* | >32 | 0.5 | 1 | 0.5 | >32/0.25 (1.5, I) | 32/0.125 (0.625, I) | 8/0.125 (0.375, S) |
| △*abc2* | >32 | 0.5 | 0.5 | 0.5 | >32/0.0625 (0.625, I) | 16/0.25 (0.75, I) | 8/0.125 (0.375, S) |
| △*abc3* | >32 | 0.5 | 0.5 | 0.125 | 16/0.125 (0.75, I) | 16/0.25 (0.75, I) | 8/0.03125 (0.375, S) |
| △*abc4* | >32 | 0.5 | 0.25 | 0.5 | 16/0.125 (0.75, I) | 16/0.125 (0.75, I) | 8/0.125 (0.375, S) |
| △*cdr1B* | >32 | 0.25 | 0.125 | 0.125 | >32/0.125 (1.5, I) | 32/0.25 (2.5, I) | 8/0.125 (1.125, I) |
| △*tca1* | >32 | 2 | 4 | 1 | 16/2 (1.25, I) | 32/4 (1.5, I) | 8/0.125 (0.25, S) |
| △*tca2* | >32 | 0.5 | 2 | 0.25 | 32/0.0625 (0.625) | 16/1 (0.75, I) | 16/0.0625 (0.5, S) |
| △*tca3* | >32 | 0.125 | 1 | 0.25 | 32/0.125 (1.5, I) | 32/0.125 (0.625, I) | 16/0.0625 (0.5, S) |
| △*tca4* | >32 | 0.5 | 0.5 | 0.5 | 16/0.0625 (0.625, I) | 16/0.25 (0.75, I) | 8/0.0625 (0.25, S) |
| △*tca5* | >32 | 0.25 | 0.25 | 0.5 | 32/0.25 (1.5, I) | 32/0.125 (1, I) | 16/0.125 (0.5, S) |
| △*tca6* | >32 | 1 | 0.25 | 0.25 | 32/0.25 (0.75, I) | 16/0.125 (0.75, I) | 8/0.0625 (0.375, S) |

[a]The MIC indicates the drug concentration suppressing growth by 100% relative to control treatment.
[b]Fractional inhibitory concentration index (FICI) values are provided in parentheses.
[c]S, synergy (FICI ≤ 0.5); I, indifference (0.5 < FICI < 4); A, antagonism (FICI ≥ 4).

significantly inhibits the activity of the efflux pump, resulting in the reduction of drug efflux (Fig. 4).

## DISCUSSION

BBM, an alkaloid derived from *Berberis amurensis* Rupr., has been the subject of prior research into its potential anticancer and anti-inflammatory properties (37, 38). These properties are thought to be mediated by the inhibition of the mitogen-activated protein kinase (MAPK) and nuclear factor kappa B (NF-κB) signaling pathways (1, 21, 39). Specifically, BBM has been demonstrated to inhibit the overexpression of cyclooxygenase-2, tumor necrosis factor-α, and NF-κB, thereby reducing the production of inflammatory factors (37).

Berberine hydrochloride (BER), an isoquinoline alkaloid that also utilizes the isoquinoline ring as a core moiety, has been demonstrated to exert a pronounced inhibitory effect on the growth of a diverse range of *Candida* species (40–42). In *A. fumigatus*, BER may inhibit fungal growth through the ergosterol biosynthesis pathway, but it does not exhibit a synergistic effect when combined with ITC. In contrast, BBM, also an isoquinoline alkaloid with the isoquinoline ring as its core structure, does not inhibit *A. fumigatus* growth when used alone but shows a strong synergistic effect when combined with POS. Currently, research on the combination of isoquinoline alkaloids and azole drugs against filamentous fungi is still limited (40). This study extends the understanding of the effects of isoquinoline alkaloids on filamentous fungi.

A total of 69 strains were examined *in vitro*. Although BBM demonstrated no antifungal activity when tested alone, the combination of BBM with POS exhibited a synergistic effect with azoles against 37 clinical isolates (54%) and 31 knockout strains (45%) of pathogenic fungi, including some clinical isolates with reduced susceptibility. A reduction in the MIC of azoles against pathogenic fungi of between four and eightfold was observed in synergistic combinations. No antagonism was observed between BBM and azoles.

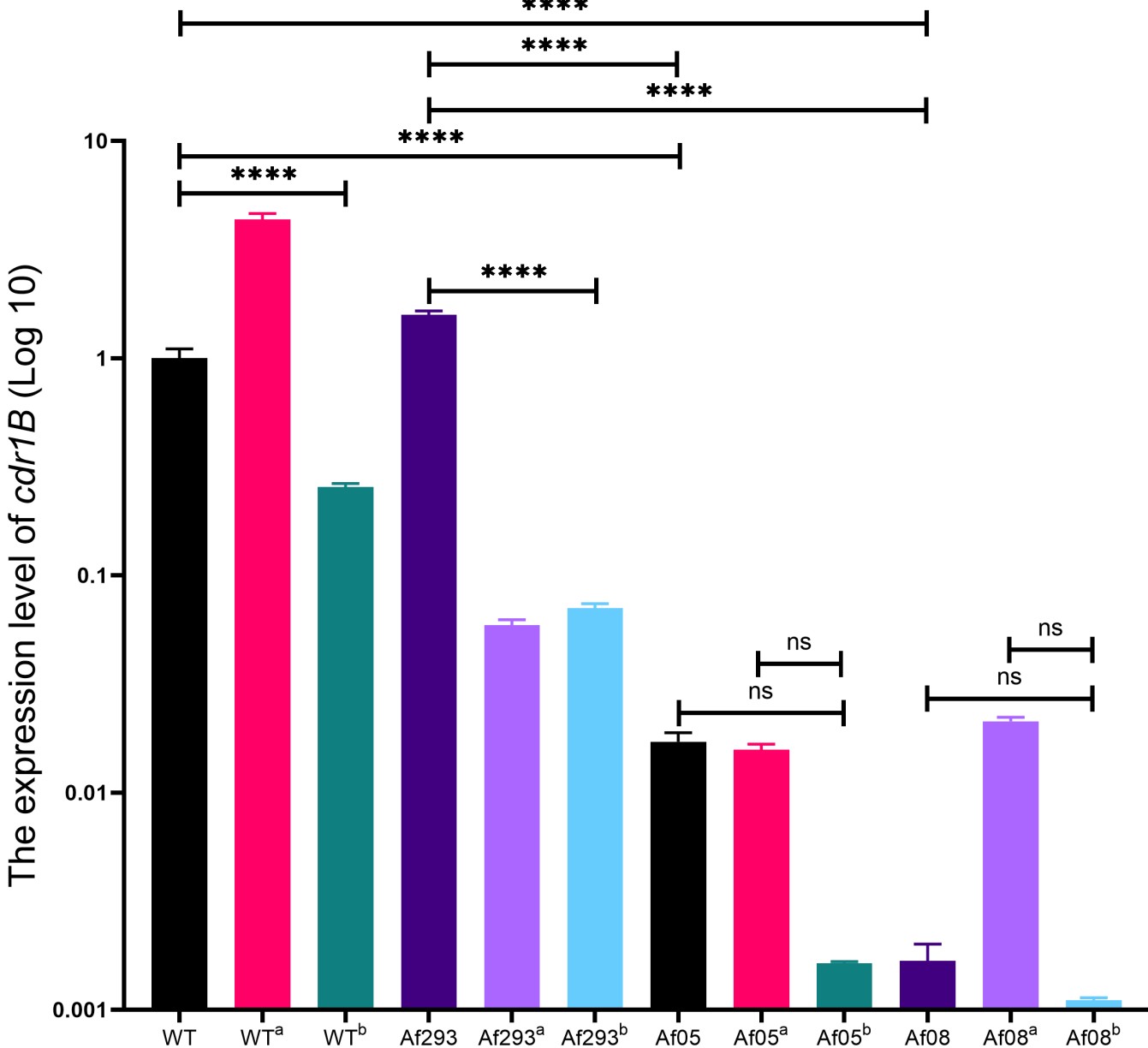

**FIG 1** Changes in *cdr1B* expression after drug stimulation. Note: [a], measurements taken after the combined stimulation with POS; [b], measurements taken after the combined stimulation with POS and BBM (****, $P < 0.0001$).

The results of the drug sensitivity experiments using the knockout strain also demonstrated that the *cdr1B* deletion strain reversed the synergistic effect with POS. This suggests that the deletion of the *cdr1B* gene may be a target of action of BBM combined with POS. In *A. fumigatus*, Cdr1B is a well-characterized ABC transporter protein. The deletion of the gene encoding this protein has been demonstrated to reverse azole resistance (43). It is hypothesized that the overexpression of MDR efflux transporter genes belonging to the ABC and MFS classes plays a significant role in the development of azole resistance in *Aspergillus* (44). In *Candida albicans*, ABC transporter genes CDR1 and CDR2, along with the major facilitator efflux gene MDR1, are implicated in the development of azole resistance (45–47). In *A. fumigatus*, the gene encoding the ABC transporter protein CDR1, AFUA_1G14330 (48), has been confirmed in this study to be the target of action of the BBM co-POS. In *A. fumigatus*, the *cdr1b* gene product is a well-characterized ABC transporter (43). This indicates that the combination of BBM and POS

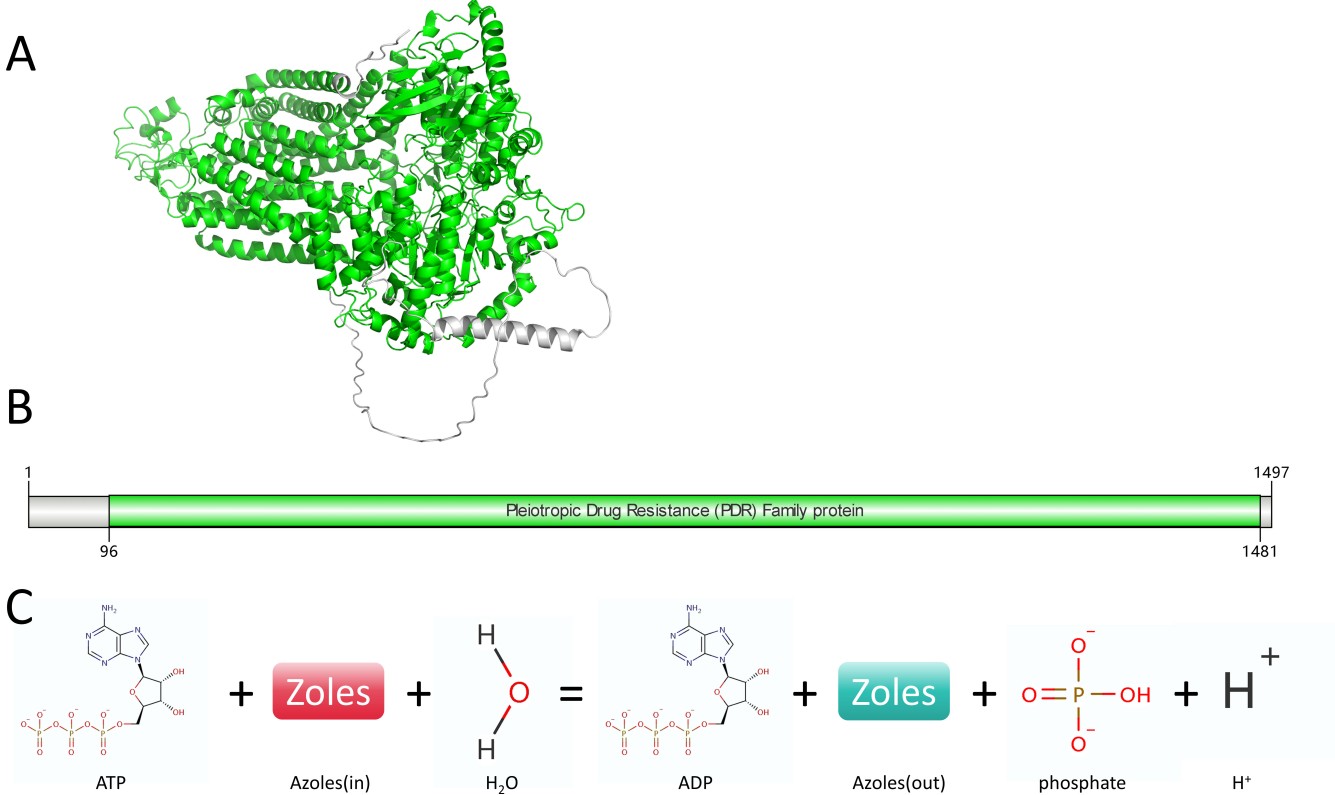

**FIG 2** Analysis of Cdr1B information. Note: the display of information related to Cdr1B. (A and B) Schematic diagrams of Cdr1B structures, with structural domains highlighted in green. (C) The process of Cdr1B pumping azole-class drugs out of the cell is also depicted.

diminishes the activity of the efflux pump, which in turn reduces drug efflux capacity. This ultimately results in a synergistic effect.

In the anti-inflammatory and anti-tumor effects of BBM, it has been demonstrated that BBM can activate the MAPK pathway and increase intracellular levels of ROS (24, 25, 49, 50). It can be inferred that changes in ROS levels are important indicators of changes in the MAPK pathway during the generative action of BBM. In this study, no further

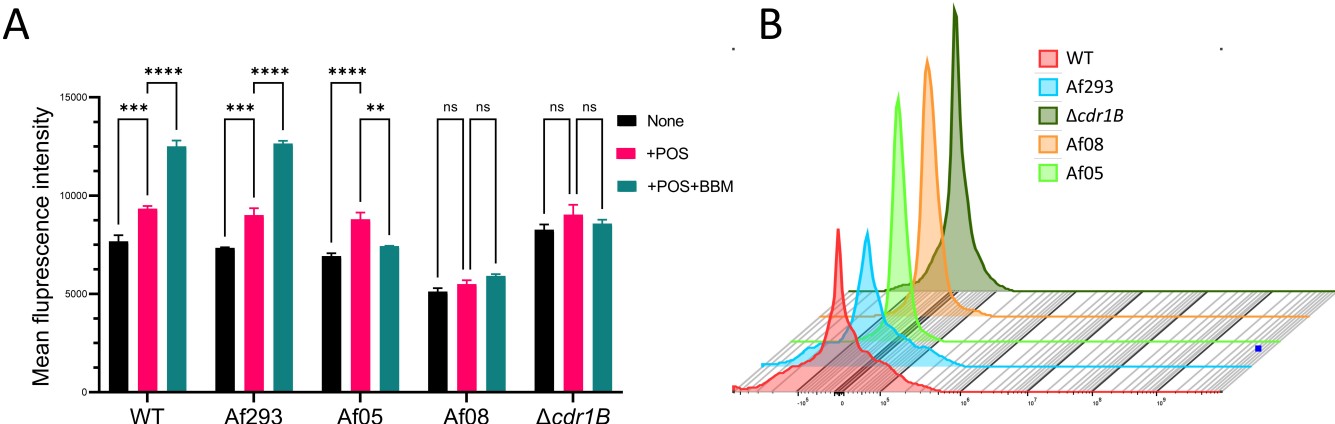

**FIG 3** ROS levels after drug stimulation. Note: (A) Each cell tagged with a fluorescent marker emits fluorescence when excited by the laser. The mean fluorescence intensity is used to quantify the average signal intensity emitted by the entire cell population. In flow cytometry, the mean value is used to represent the level of ROS within the cells. This method allows researchers to accurately assess the relative abundance of ROS within the cells, thereby understanding the cell's response to oxidative stress. (B) A visual representation of ROS levels after the combined stimulation with POS and BBM (**$P < 0.01$, ***$P < 0.001$, and ****$P < 0.0001$).

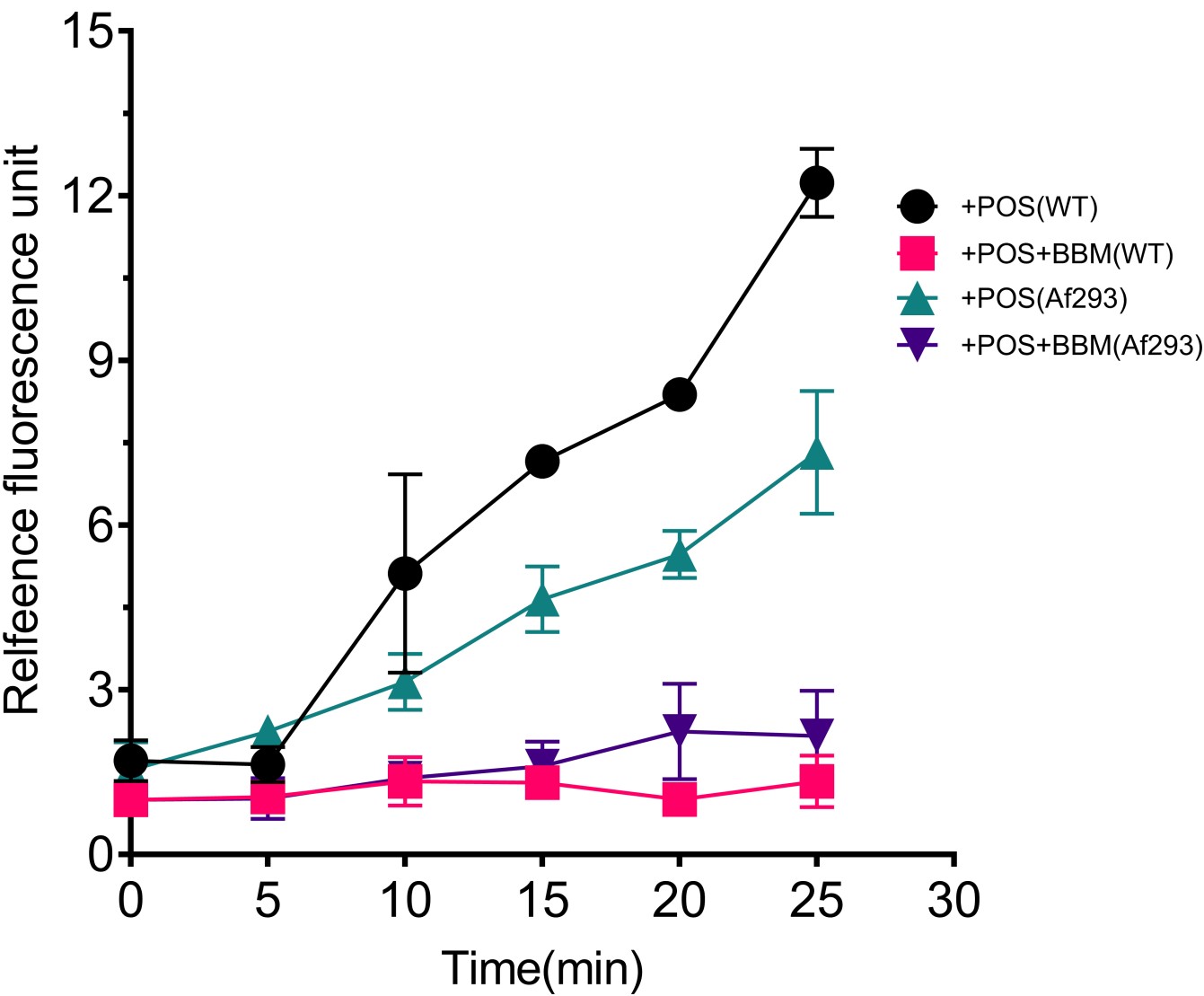

**FIG 4** Influence of BBM combined with POS on R6G exhaust. Note: energy-dependent Cdr1B-mediated efflux of R6G from yeast cells. Deenergized cells were preloaded with R6G as described in "Materials and Methods." The efflux of rhodamine was followed by direct measurement of the fluorescence in cell supernatants following the addition of glucose (2 mM) to suspensions of WT and Af293. The groups stimulated with the combination of two drugs (POS + BBM) were normalized, and the ratio of the group stimulated with the single drug (POS) to the group stimulated with both drugs was considered as relative fluorescence intensity. It can be observed that after the addition of BBM, the efflux of Rhodamine significantly decreased, indicating a significant inhibition of Cdr1B activity under the combined drug stimulation.

increase in ROS was observed in strains without synergism, whereas ROS levels were increased in strains with synergism after the addition of BBM. MAPKs are conserved from yeast to humans and play relevant roles in cellular physiology (51). In fungal pathogens, the conserved MAPK pathway controls key virulence functions. The alterations in ROS levels observed in this study may also be attributed to modifications in the MAPK pathway in response to BBM stimulation (52–54). Nevertheless, further research is necessary to elucidate the underlying mechanism and to ascertain the potential reliability and safety of their clinical application.

## Conclusions

This study demonstrates that BBM combined with POS significantly enhances antifungal activity against *A. fumigatus*. This synergistic effect may be achieved by inhibiting efflux pump activity and increasing mitochondrial ROS production. The combination of BBM

and POS offers a potentially effective therapeutic strategy to address azole resistance in the treatment of invasive fungal infections.

## ACKNOWLEDGMENTS

We thank everyone who contributed to the success of this research, including colleagues, institutions, and funding bodies.

This work was supported by the Jingzhou Science and Technology Plan Project (grant numbers 2024HD34); the Open Research Fund Program of the State Key Laboratory of Virology of China (grant numbers 2023KF006); the Yangtze University Science and Technology Aid to Tibet Medical Talent Training Program Project (grant numbers 2023YZ06); and the Key Research and Development program of Hubei Province (grant numbers 2024BCB043).

All authors contributed to the research in this report. Individual contributions are as follows: Y.S. designed the study; H.Z. and Y.Z.Z. performed the experiments; H.M.Y. and X.Y.T. developed and optimized the model and wrote the manuscript; F.Q.D. interpreted data and carried out statistical analyses; Y.S. made final critical revision for important intellectual contents. All authors approved the final version of the paper.

## AUTHOR AFFILIATIONS

[1]Department of Dermatology, Jingzhou Hospital Affiliated to Yangtze University, Hubei Provincial Clinical Research Center for Diagnosis and Therapeutics of Pathogenic Fungal Infection, Jingzhou, Hubei Province, China
[2]Department of Clinical Laboratory, Jingzhou Hospital Affiliated to Yangtze University, Hubei Provincial Clinical Research Center for Diagnosis and Therapeutics of Pathogenic Fungal Infection, Jingzhou, Hubei Province, China
[3]Department of Clinical Medical, Yangtze University, Jingzhou, Hubei, China
[4]Department of Clinical Medical, People's Hospital of Naidong District, Shannan, China

## AUTHOR ORCIDs

Heng Zhang  http://orcid.org/0009-0002-6618-5311
Yi Sun  http://orcid.org/0000-0002-4489-3803

## FUNDING

| Funder | Grant(s) | Author(s) |
| --- | --- | --- |
| Jingzhou Science and Technology Plan Project | 2024HD34 | Heng Zhang |
| Open Research Fund Program of the State Key Laboratory of Virology of China | 2023KF006 | Yizheng Zhou |
| Yangtze University Science and Technology Aid to Tibet Medicial Talent Training Program Project | 2023YZ06 | Yi Sun |
| Hebei Provincial Key Research Projects (Key Research and Development Program of Hebei Province) | 2024BCB043 | Yi Sun |

## AUTHOR CONTRIBUTIONS

Heng Zhang, Funding acquisition, Methodology, Resources, Software, Writing – original draft | Yizheng Zhou, Funding acquisition, Investigation, Software | Hanmo Yang, Formal analysis, Methodology, Project administration | Xinyi Tao, Software, Validation, Writing – original draft | Yinping Chen, Methodology, Project administration | Fuqiang Dong, Investigation, Visualization | Yi Sun, Conceptualization, Formal analysis, Funding acquisition, Investigation, Resources, Software, Validation, Visualization

## DATA AVAILABILITY

The data sets used and/or analyzed during the current study are available from the corresponding author on reasonable request.

## ETHICS APPROVAL

This study was performed in line with the principles of the Declaration of Helsinki. Approval was granted by the Ethics Committee of Jingzhou Hospital Afliated to Yangtze University.

## ADDITIONAL FILES

The following material is available online.

### Supplemental Material

**Supplemental material (Spectrum03184-24-s0001.docx).** (i) Generation of deletion mutants. (ii) Strain identification. (iii) Gene expression analysis. Figure S1: Schematic diagram of the chemical structure of berbamine hydrochloride. Figure S2: Schematic overview of fusion PCR-based generation of gene knockout mutants in *A. fumigatus*. Table S1: Information on missing genes in gene-deficient strains. Table S2: Primer sets and corresponding amplification targets. Table S3: Real-time quantitative PCR primer list. Table S4: Primer list of generation of deletion mutants.

### Open Peer Review

**PEER REVIEW HISTORY (review-history.pdf).** An accounting of the reviewer comments and feedback.

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
