## [Reviewer comments · Microbiology Spectrum]

Microbiology Spectrum

In vitro* Interactions of Berbamine Hydrochloride and Azoles against *Aspergillus fumigatus

Heng Zhang, Yizheng Zhou, Hanmo Yang, Xinyi Tao, Yiping Chen, Fuqiang Dong, and Yi Sun

Corresponding Author(s): Yi Sun, Jingzhou Hospital Affiliated to Yangtze University

Review Timeline:

Submission Date:	December 6, 2024
Editorial Decision:	January 21, 2025
Revision Received:	February 11, 2025
Accepted:	February 26, 2025

Editor: Vera Tesic

Reviewer(s): Disclosure of reviewer identity is with reference to reviewer comments included in decision letter(s). The following individuals involved in review of your submission have agreed to reveal their identity: Payal Gupta (Reviewer #1)

Transaction Report:

DOI: <https://doi.org/10.1128/spectrum.03184-24>

Re: Spectrum03184-24 (*In vitro* Interactions of Berbamine Hydrochloride and Azoles against *Aspergillus fumigatus*)

Dear Dr. Yi Sun:

Thank you for the privilege of reviewing your work. Below you will find my comments, instructions from the Spectrum editorial office, and the reviewer comments.

Revision Guidelines

Sincerely,
Vera Tesic
Editor
Microbiology Spectrum

Reviewer #1 (Comments for the Author):

The research article entitled "In vitro Interactions of Berbamine Hydrochloride and Azoles against *Aspergillus fumigatus*" by Zhang et al. has investigated the effect of berberine hydrochloride in combination with azoles against *Aspergillus*. The authors have done a complete, focused, and impactful study that has restored the therapeutic efficacy of the azoles in the co-administration of BBM. Before recommending the manuscript for publication in the journal, I have the following suggestions.

1. Line 40: correct spelling of BBM.
2. Correct the scientific writing of gene in the manuscript.
3. The effectivity of berberine is well reported in fungal strains. So how the effectiveness of BBM is different from it. If there are differences, please add in discussion.
4. Why authors have used lab strain of *C. parapsilosis* with *Aspergillus* as quality control?
5. Mention percentage of DMSO used for drug dilution preparation.
6. Detailed methodology used for creating knockout mutants.
7. Why authors have not mentioned anything about MFS transporters expression and role?
8. Authors must mention the important insight of their study about efflux pump involvement in this synergistic study in the abstract.

Reviewer #2 (Comments for the Author):

Spectrum03184-24

Aspergillus fumigatus is an environmental fungus and the leading cause of invasive aspergillosis in immunocompromised individuals, with a mortality rate reaching up to 90%, depending on the severity of the infection. While azoles are commonly used to treat this condition, resistance in certain strains has limited treatment options. Therefore, the evaluation of new antifungal agents or natural compounds to treat aspergillosis is crucial. In this study, the authors examined the synergistic effect of berbamine hydrochloride, a natural compound, in combination with azoles to treat *A. fumigatus*, and also investigated the underlying mechanisms involved. Below are my comments.

Major comments:

Line 25: When performing in vitro susceptibility testing, it is essential to confirm the identity of the isolates and specify the guidelines followed for performing antifungal susceptibility testing and interpreting the MIC results.

Line 57-58: You've provided the efficacy of voriconazole and amphotericin. Also include efficacy of the other azole drugs you've tested e.g. posaconazole and itraconazole, especially since your results indicated that when BBM was combined with posaconazole it was more effective.

Line 78-80: Although this sentence indicates the study's objective, it is important to explicitly state the aim of the study here. For example, you could clarify that "the aim of this study was to..."

Line 86: When performing AFST, it is recommended to include both susceptible and resistant *Aspergillus* ATCC strains as controls. Can you clarify why a yeast strain was used in this case, and during which part of the study these controls were applied? Was it during AFST or molecular testing?

Line 87-91: This section is overly summarized. To ensure the experiment is reproducible by other researchers, please provide detailed procedures for each method in the supplementary materials.

Line 94: Please elaborate on how AFST was conducted for each azole drug, and specify the guidelines used for interpreting the MIC results.

Line 110-125: This section requires rewriting for better clarity and readability.

Line 150-151: For statistical analysis, ANOVA was used to compare what? Please clarify the comparison. Also, rephrase the sentence regarding statistical significance.

Line 152: You performed three PCR assays solely for species identification. Please summarize the results of these assays and include them in lines 152-153.

Line 155: Consider including a table that shows the MIC distribution for each antifungal agent without the combination treatments. For example, include the number of isolates with an MIC of 0.25 for posaconazole, etc.

Line 156-161: Specify the number of isolates for each group, e.g., clinical isolates (n=36), defective strains (n=31), MFS transport strains (n=20), etc. This will make the data easier to understand.

Line 178: You are reporting individual isolate data here; please also include the results for the three groups tested.

Line 179: Explain why *cdr1B* is likely a target for the BBM-posaconazole combination rather than BBM-itraconazole or BBM-voriconazole.

Line 204: Again, summarize the results for the three groups first. This will provide context for your focus on the BBM- posaconazole combination over the other drug combinations.

Line 285-290 and 292-295: These sections repeat the same content. Please select one version to include in the manuscript as the conclusion.

Line 298: Justify why ethical approval was not obtained, considering that clinical isolates were used in the study.

Minor comments:

Line 30: Where do the 66 strains and defective strains come from? Are they also clinical strains, or were they isolated from the environment?

Line 32: When you refer to "these resistant strains," which specific strains are you talking about?

Line 32-33: Under the methods section, please specify the number and types of the 69 *A. fumigatus* strains tested, including whether they are wild-type strains.

Line 63: Consider moving Figure 1 to the supplementary section.

Line 73: Clarify whether berbamine hydrochloride (BBM) was effective when used alone in treating cancer, or only when combined with other agents.

Line 83: What is a standard strain, and how does it differ from the others?

Line 84: Please explain what constitutes a standard strain. Aren't the wild-type and genetically deficient strains also isolated from patients? If these are environmental strains, please mention this as well.

Line 90-91: Rephrase to clarify that sequencing of the beta-tubulin and calmodulin genes was performed to confirm species identity. Also, provide detailed procedures in the supplementary section.

Line 93: Please write out the full name of VOR.

Line 94: Include the company and country of purchase for the drugs.

Line 96: Is it SDA or SAB? What does the "B" in SAB stand for? Please ensure this is corrected throughout the manuscript.

Line 101: Specify where "As described"

Line 115: What did the *A. fumigatus* control group receive during the experiment?

Line 114-115: Rephrase "culturing at 37°C." Do you mean that the isolates were cultured and incubated at 37°C?

Line 127: Suggest you remove "refer to 2.4 for grouping" and instead add "for each group" at the end of the sentence in line 129.

Line 127: Change to TRleasy.

Line 128: Add "as per the manufacturer's recommendation."

Line 129: Provide more details about the method in the supplementary section.

Line 132: What are the "mutual control groups" referred to here?

Line 134: Correct "fumigates" to *fumigatus*.

Line 135: Suggest you remove "refer to 2.4 for grouping" and instead add "for each group" at the end of the sentence.

Line 135: Again, correct "fumigates" to *fumigatus*.

Line 146: The microplate reader was read at the excitation wavelength of...

Line 180: Include, from the 36 clinical isolates, Af05...

Line 183: After Af293, include "(standard strain)" for clarification.

Line 192-202: I would suggest moving this information to the supplementary section.

Line 255: Why have you separated these isolates into pathogenic and knockout categories? Does this mean that since these strains have been knocked out, they are no longer pathogenic? Are the genes in question responsible for *A. fumigatus* virulence or antimicrobial resistance?

Microbiology Spectrum

In vitro Interactions of Berbamine Hydrochloride and Azoles against *Aspergillus fumigatus*

The research article entitled “**In vitro Interactions of Berbamine Hydrochloride and Azoles against *Aspergillus fumigatus***” by Zhang et al have investigated the effect of berbamine hydrochloride in combination with azoles against *Aspergillus*. Authors have done a complete, focussed and impactful study which has restored the therapeutic efficacy of the azoles in co-administration of BBM. I have following suggestion for before recommending manuscript for publication in the journal.

1. Line 40: correct spelling of BBM.
2. Correct the scientific writing of gene in the manuscript.
3. The effectivity of berberine is well reported in fungal strains. So how the effectiveness of BBM is different from it. If there are differences, please add in discussion.
4. Why authors have used lab strain of *C. parapsilosis* with *Aspergillus* as quality control?
5. Mention percentage of DMSO used for drug dilution preparation.
6. Detailed methodology used for creating knockout mutants.
7. Why authors have not mentioned anything about MFS transporters expression and role?
8. Authors must mention the important insight of their study about efflux pump involvement in this synergistic study in the abstract.

Spectrum03184-24

Aspergillus fumigatus is an environmental fungus and the leading cause of invasive aspergillosis in immunocompromised individuals, with a mortality rate reaching up to 90%, depending on the severity of the infection. While azoles are commonly used to treat this condition, resistance in certain strains has limited treatment options. Therefore, the evaluation of new antifungal agents or natural compounds to treat aspergillosis is crucial. In this study, the authors examined the synergistic effect of berbamine hydrochloride, a natural compound, in combination with azoles to treat *A. fumigatus*, and also investigated the underlying mechanisms involved. Below are my comments.

Major comments:

Line 25: When performing in vitro susceptibility testing, it is essential to confirm the identity of the isolates and specify the guidelines followed for performing antifungal susceptibility testing and interpreting the MIC results.

Line 57-58: You've provided the efficacy of voriconazole and amphotericin. Also include efficacy of the other azole drugs you've tested e.g. posaconazole and itraconazole, especially since your results indicated that when BBM was combined with posaconazole it was more effective.

Line 78-80: Although this sentence indicates the study's objective, it is important to explicitly state the aim of the study here. For example, you could clarify that "the aim of this study was to..."

Line 86: When performing AFST, it is recommended to include both susceptible and resistant *Aspergillus* ATCC strains as controls. Can you clarify why a yeast strain was used in this case, and during which part of the study these controls were applied? Was it during AFST or molecular testing?

Line 87-91: This section is overly summarized. To ensure the experiment is reproducible by other researchers, please provide detailed procedures for each method in the supplementary materials.

Line 94: Please elaborate on how AFST was conducted for each azole drug, and specify the guidelines used for interpreting the MIC results.

Line 110-125: This section requires rewriting for better clarity and readability.

Line 150-151: For statistical analysis, ANOVA was used to compare what? Please clarify the comparison. Also, rephrase the sentence regarding statistical significance.

Line 152: You performed three PCR assays solely for species identification. Please summarize the results of these assays and include them in lines 152-153.

Line 155: Consider including a table that shows the MIC distribution for each antifungal agent without the combination treatments. For example, include the number of isolates with an MIC of 0.25 for posaconazole, etc.

Line 156-161: Specify the number of isolates for each group, e.g., clinical isolates (n=36), defective strains (n=31), MFS transport strains (n=20), etc. This will make the data easier to understand.

Line 178: You are reporting individual isolate data here; please also include the results for the three groups tested.

Line 179: Explain why *cdr1B* is likely a target for the BBM-posaconazole combination rather than BBM-itraconazole or BBM-voriconazole.

Line 204: Again, summarize the results for the three groups first. This will provide context for your focus on the BBM-posaconazole combination over the other drug combinations.

Line 285-290 and 292-295: These sections repeat the same content. Please select one version to include in the manuscript as the conclusion.

Line 298: Justify why ethical approval was not obtained, considering that clinical isolates were used in the study.

Minor comments:

Line 30: Where do the 66 strains and defective strains come from? Are they also clinical strains, or were they isolated from the environment?

Line 32: When you refer to “these resistant strains,” which specific strains are you talking about?

Line 32-33: Under the methods section, please specify the number and types of the 69 *A. fumigatus* strains tested, including whether they are wild-type strains.

Line 63: Consider moving Figure 1 to the supplementary section.

Line 73: Clarify whether berbamine hydrochloride (BBM) was effective when used alone in treating cancer, or only when combined with other agents.

Line 83: What is a standard strain, and how does it differ from the others?

Line 84: Please explain what constitutes a standard strain. Aren't the wild-type and genetically deficient strains also isolated from patients? If these are environmental strains, please mention this as well.

Line 90-91: Rephrase to clarify that sequencing of the beta-tubulin and calmodulin genes was performed to confirm species identity. Also, provide detailed procedures in the supplementary section.

Line 93: Please write out the full name of VOR.

Line 94: Include the company and country of purchase for the drugs.

Line 96: Is it SDA or SAB? What does the “B” in SAB stand for? Please ensure this is corrected throughout the manuscript.

Line 101: Specify where “As described”

Line 115: What did the *A. fumigatus* control group receive during the experiment?

Line 114-115: Rephrase “culturing at 37°C.” Do you mean that the isolates were cultured and incubated at 37°C?

Line 127: Suggest you remove “refer to 2.4 for grouping” and instead add “for each group” at the end of the sentence in line 129.

Line 127: Change to TRleasy.

Line 128: Add "as per the manufacturer's recommendation."

Line 129: Provide more details about the method in the supplementary section.

Line 132: What are the "mutual control groups" referred to here?

Line 134: Correct "fumigates" to *fumigatus*.

Line 135: Suggest you remove "refer to 2.4 for grouping" and instead add "for each group" at the end of the sentence.

Line 135: Again, correct "fumigates" to *fumigatus*.

Line 146: The microplate reader was read at the excitation wavelength of...

Line 180: Include, from the 36 clinical isolates, Af05...

Line 183: After Af293, include "(standard strain)" for clarification.

Line 192-202: I would suggest moving this information to the supplementary section.

Line 255: Why have you separated these isolates into pathogenic and knockout categories? Does this mean that since these strains have been knocked out, they are no longer pathogenic? Are the genes in question responsible for *A. fumigatus* virulence or antimicrobial resistance?

Dear Editor,

We are very grateful for your constructive comments and suggestions for our manuscript entitled "*In vitro* Interactions of Berbamine Hydrochloride and Azoles against *Aspergillus fumigatus*"(Spectrum03184-24). Your comments are very valuable and helpful for improving our manuscript. In the following,the responses to all the comments are provided one by one.

Reviewer #1 (Comments for the Author):

1. Line 40: correct spelling of BBM.

Answer: Thank you for your correction. I have changed "berbaminee hydrochloride" to "berbamine hydrochloride".

2. Correct the scientific writing of gene in the manuscript.

Answer: Thank you for your feedback. I have corrected the spelling of the genes in the manuscript.

3. The effectivity of berberine is well reported in fungal strains. So how the effectiveness of BBM is different from it. If there are differences, please add in discussion.

Answer: Thank you for your feedback. I have added content to the discussion section:

Berberine hydrochloride (BER), an isoquinoline alkaloid that also utilises the isoquinoline ring as a core moiety, has been demonstrated to exert a pronounced inhibitory effect on the growth of a diverse range of *Candida* species[40-42], In *A.fumigatus*, BER may inhibit fungal growth through the ergosterol biosynthesis pathway , but it does not exhibit a synergistic effect when combined with ITC. In contrast, BBM, also an isoquinoline alkaloid with the isoquinoline ring as its core structure, does not inhibit *A. fumigatus* growth when used alone, but shows a strong synergistic effect when combined with POS. Currently, research on the combination of isoquinoline alkaloids and azole drugs against filamentous fungi is still limited[40].This study extends the understanding of the effects of isoquinoline alkaloids on filamentous fungi.

4. Why authors have used lab strain of *C. parapsilosis* with *Aspergillus* as quality control?

Answer: *Candida parapsilosis* (ATCC 22019) and *Aspergillus flavus* (ATCC 204304) are

commonly used quality control strains in antifungal susceptibility testing. The MIC values for these strains are fixed, and by measuring the MIC values in specific experiments, we can determine whether the antifungal susceptibility plates used in the experiment meet the required standards. In this study, dual quality control was used to ensure the accuracy of the results.

5. Mention percentage of DMSO used for drug dilution preparation.

Answer: Thank you for your feedback. In this experiment, the stock concentration of the drugs was 6400 µg/mL, and DMSO (purity≥99.9%, Aladdin, China) was used for dilution. In the antifungal susceptibility testing, the concentration of DMSO did not exceed 0.5%. I have added this information in the manuscript.

6. Detailed methodology used for creating knockout mutants.

Answer: Thank you for your suggestion. I have provided a detailed description of the methodology for creating knockout mutants in the supplementary file:

1. Generation of deletion mutants

A. fumigatus AF293 (Fungal Genetics Stock Center), used as the parent strain for all amplification in this investigation, includes the whole DNA sequence of *A. fumigatus*. *A. fumigatus* A1160 (Δ KU80, *pyrG*⁻) (Fungal Genetics Stock Center) is a uracil (U)-deficient strain, defective in the *pyrG* gene, and unable to grow on U-deficient media.

1.1 Preparation of Knockout Product: The target gene sequence was retrieved from the NCBI database, and primers P1+P2 and P3+P4 were designed for the target gene. Genomic DNA of AF293 was used as a template to amplify 1200 bp sequences upstream and downstream of the target gene, obtaining the flanking regions. The plasmid pBARGPE1-PyrG-TagRFP (Wensheng Biotechnology, Hunan, China) DNA was extracted as the amplification template, and primers PyrG-F and PyrG-R were used to amplify the *pyrG* gene sequence, obtaining the *pyrG* fragment. Finally, using primers P5+P6, along with the upstream and downstream fragments and the *pyrG* fragment, PCR amplification was conducted to obtain the complete knockout product (Fig.S2, Table S4).

1.2 Preparation of Protoplasts: 150 µL of A1160 spores (1×10^9 cfu/mL) were inoculated into 50 mL of SAB liquid medium containing uracil (0.05 g/100 mL) in a sterile 50 mL flask, which was

sealed with sterile foam and covered with sterile aluminum foil. The flask was incubated at 37°C and 130 rpm for 16 hours. After spore growth, mycelium was collected using miracloth (Millipore Sigma) and transferred to a new sterile 250 mL conical flask. Pre-prepared protoplast solution[1] was added, and the mixture was incubated on a shaking incubator at 37°C, 100 rpm/min for 4 to 5 hours until most spores had transformed into protoplasts. Miracloth was used to remove incompletely digested mycelium.

1.3 The filtered solution was transferred to a 50 mL conical tube and centrifuged (1800 g, 4°C, 10 minutes). After removing the supernatant, the pellet was resuspended in 2 mL of KCl/CaCl₂ solution and divided into two sterile 1.5 mL EP tubes. The suspension was further centrifuged at 4°C, 900 g for 3 minutes, and the supernatant discarded. This process was repeated twice, and the protoplasts from both tubes were combined into one tube. A final centrifugation (1800 g, 4°C, 3 minutes) was performed, and the supernatant was discarded. A 1 µL aliquot of the suspension was placed on a hemocytometer, and the concentration was determined using a microscope. The final concentration was adjusted to 1×10^7 cfu/mL, and the final protoplast solution was prepared by mixing with 1 mL of KCl/CaCl₂ solution.

1.4 Transformation Procedure: 20 µL of the fusion PCR product was added to a sterile EP tube, followed by 20 µL of filtered PEG solution and 50 µL of the prepared protoplast suspension. The mixture was gently pipetted and incubated on ice for 30 minutes. An additional 100 µL of filtered PEG solution was added, and the mixture was pipetted again and incubated on ice for 5 minutes to allow the fusion product, which carries the marker gene, to successfully enter the protoplasts and undergo homologous recombination.

1.5 The mixture was plated onto CZA solid medium without uracil and incubated at room temperature for 24 hours. Afterward, the plate was transferred to a 37°C incubator and cultured for 3 to 5 days, and transformants were selected.

Fig.S2 Schematic overview of fusion PCR based generation of gene knockout mutants in *A. fumigatus*

Note: (A) Upstream and downstream fragments are amplified with primers P1 and P2 and with P3 and P4. The *pyrG* selective marker cassette is amplified with primers PyrG-F and PyrG-R. (B) The upstream and downstream fragments are fused to the *pyrG* selective cassette by fusion PCR using nested primers (P5 and P6) creating a linear fragment suitable for transformation. (C) The final knockout product generated.

7. Why authors have not mentioned anything about MFS transporters expression and role?

Answer: Thank you for your feedback. I have added information regarding the expression and role of MFS transporters in the introduction section:

These pumps are involved in the active extrusion of antimicrobial molecules, including azoles. MDR efflux transporter genes from both the ATP-binding cassette (ABC) and major facilitator superfamily (MFS) classes have been shown to play a clinically significant role in various pathogenic fungi[9, 10]. Sequence analysis suggests that *A. fumigatus* contains 278 different MFS and 49 ABC transporters[11]. Several studies have described the role of *A. fumigatus* MDR (AfuMDR) pumps, showing their association with increased resistance to ITC[12, 13]. In conclusion, these MDR efflux pumps influence the sensitivity of the fungus to azoles.

8. Authors must mention the important insight of their study about efflux pump involvement in this synergistic study in the abstract.

Answer: Thank you for your suggestion. I have emphasized the involvement of efflux pumps as

an important factor in the synergy in the abstract, as follows:

Objective: The growing resistance of *Aspergillus* to azoles poses a significant challenge in treating invasive fungal infections. This study aimed to investigate the synergistic effect of berbaminee hydrochloride (BBM) combined with azoles in treating *Aspergillus fumigatus* and explore the role of efflux pump inhibition in this synergy.

Methods: The efficacy of combining BBM with itraconazole (ITC), voriconazole (VOR), and posaconazole (POS) was tested against 69 *A. fumigatus* strains that had been identified using the M38-A3 broth microdilution method. RT-qPCR was used to measure gene expression related to synergy, while flow cytometry was employed to assess mitochondrial ROS levels, and Rhodamine 6G exocytosis assays were performed to quantify efflux pump activity.

Results: BBM alone showed no significant antifungal activity. BBM combined with POS exhibited synergy against 66 strains (95.7%), while two clinical isolates (Af05/Af08) and one defective strain ($\Delta cdr1B$) showed no synergy. Synergy with ITC was observed in 3 strains (4.3%), and none with VOR. In the non-synergistic Af05 and Af08 strains, the expression of the *cdr1B* gene was significantly lower compared to wild-type (WT) strains. ROS levels increased significantly in WT with POS and BBM combination therapy, but not in the defective strains. Glucose uptake was also reduced in the POS-BBM combination.

Conclusions: BBM enhances azole sensitivity in *A. fumigatus* primarily by inhibiting the *cdr1B*-mediated efflux pump, supported by reduced Rhodamine 6G exocytosis and synergy loss in *cdr1B*-deficient strains. ROS accumulation and metabolic disruption may further contribute to this synergy. Targeting efflux pumps with BBM provides a novel strategy to combat azole resistance.

Reviewer #2 (Comments for the Author):

Major comments:

Line 25: When performing in vitro susceptibility testing, it is essential to confirm the identity of the isolates and specify the guidelines followed for performing antifungal susceptibility testing and interpreting the MIC results.

Answer: Thank you for your feedback. Prior to the antifungal susceptibility testing, the strains in this study were first identified by extracting DNA from the cultured strains and performing PCR amplification using the ITS1 and ITS4, cmd5 and cmd6, Bt2a and Bt2b, followed by sequencing. The sequencing results were compared against the NCBI database to confirm the species. The antifungal susceptibility testing in this study was performed according to the M38-A3 broth microdilution method. I have also revised this section as follows:

Methods: The efficacy of combining BBM with itraconazole (ITC), voriconazole (VOR), and posaconazole (POS) was tested against 69 *A. fumigatus* strains that had been identified using the M38-A3 broth microdilution method. RT-qPCR was used to measure gene expression related to synergy, while flow cytometry was employed to assess mitochondrial ROS levels, and Rhodamine 6G exocytosis assays were performed to quantify efflux pump activity.

Line 57-58: You've provided the efficacy of voriconazole and amphotericin. Also include efficacy of the other azole drugs you've tested e.g. posaconazole and itraconazole, especially since your results indicated that when BBM was combined with posaconazole it was more effective.

Answer: Thank you for your feedback. I have added the following information:

In addition, itraconazole(ITC) has an efficacy of 43-76%, and posaconazole(POS) has an efficacy of 44-61%[7]. It is worth noting that the treatment rate for invasive fungal infections in hematopoietic stem cell transplant patients with POS is 57.8%[8].

Line 78-80: Although this sentence indicates the study's objective, it is important to explicitly state the aim of the study here. For example, you could clarify that "the aim of this study was to..."

Answer: Thank you for your feedback. I have modified this part as follows: The aim of this study was to investigate the synergistic effect of BBM combined with azole drugs against *A. fumigatus* *in vitro*, providing a potential high-efficiency strategy for combating IA.

Line 86: When performing AFST, it is recommended to include both susceptible and resistant

Aspergillus ATCC strains as controls. Can you clarify why a yeast strain was used in this case, and during which part of the study these controls were applied? Was it during AFST or molecular testing?

Answer: *Candida parapsilosis* (ATCC 22019) and *Aspergillus flavus* (ATCC 204304) are commonly used quality control strains in antifungal susceptibility testing, as their MIC values for individual drugs are fixed. By measuring the MIC values in specific experiments, it is possible to determine whether the antifungal susceptibility plates prepared for the experiment meet the required standards. Specifically, these controls were applied during the AFST.

Line 87-91: This section is overly summarized. To ensure the experiment is reproducible by other researchers, please provide detailed procedures for each method in the supplementary materials.

Answer: Thank you for your suggestion. I have provided the detailed procedures for each method in the supplementary materials, specifically in section **1. Generation of deletion mutants**.

Line 94: Please elaborate on how AFST was conducted for each azole drug, and specify the guidelines used for interpreting the MIC results.

Answer: Thank you for your feedback. I have made the following revision in section **2.3. Testing the *in vitro* Synergy of BBM and Azoles**

Susceptibility testing was performed according to the broth microdilution chequerboard procedure based on the CLSI M27-A3[30] standard and previously published protocols[31, 32]. Conidia harvested from cultures grown for 3 days on SAB solid medium were suspended in sterile distilled water containing 0.03% Triton and diluted to a concentration of $2-5 \times 10^6$ spores/mL, which were then diluted 100 times in RPMI 1640 to achieve a 2-fold suspension more concentrated than the density needed or to approximately $1-3 \times 10^4$ spores/mL[32]. The working concentration ranges of BBM, ITR, VRC and POS were 0.5–32 µg/mL, 0.06–8 µg/mL, 0.06–8 µg/mL and 0.03–4 µg/mL, respectively. A 50 µL of BBM with serial dilutions were inoculated in horizontal direction and another 50 µL of azoles with serial dilutions were inoculated in vertical direction on the 96-well plate, which contained 100 µL prepared inoculum suspension. Interpretation of results was performed after incubation at 35°C for 48h. The MICs applied for the evaluation of effects against *A. fumigatus* was determined as the

lowest concentration resulting in 100% inhibition of growth[32].The FICI as calculated by the formula: $FICI=(Ac/Aa)+(Bc/Ba)$, where Ac and Bc are the MICs of antifungal drugs in combination, and Aa and Ba are the MICs of antifungal drugs A and B alone[33] . A FICI of ≤ 0.5 is classified as synergy, a FICI of >0.5 to ≤ 4 indicates no interaction (indifference), and a FICI of >4 indicates antagonism[34]. All tests were performed in triplicate

Line 110-125: This section requires rewriting for better clarity and readability.

Answer: Thank you for your feedback. I have revised this section as follows:

2.4. Detection of ROS content

A. fumigatus conidia were collected after 48 hours of incubation on SAB solid medium at 35°C and resuspended in 1640 liquid medium to a final concentration of 5×10^4 cfu/mL. All samples were incubated at 37°C in a shaking incubator at 130 rpm for 16 hours. During the incubation period, drug treatments were applied as needed. At the end of the incubation, 2,7-dichlorodihydrofluorescein diacetate (DCFH-DA, Yeasen Biotechnology, Shanghai, China) was added to each sample and incubated at 37°C for 1 hour to detect intracellular ROS levels. Flow cytometry data acquisition was performed using a Beckman Cytomics FC 500 BD FACSCanto II, and data analysis was conducted using FlowJo v10 software. The excitation wavelength was set at 488 nm, and the emission wavelength at 525 nm.

Line 150-151: For statistical analysis, ANOVA was used to compare what? Please clarify the comparison. Also, rephrase the sentence regarding statistical significance.

Answer: Thank you for your feedback. The statistical analysis was used to compare differences between groups. I have revised this section as follows: The GraphPad Prism 8.0 was used for statistical analysis. The results were presented as mean \pm SD or mean \pm SEM. Student t-test, One-Way ANOVA, or Two-Way ANOVA were employed for statistical comparison. Statistical significance was determined with 95% (*, $P < 0.05$), 99% (**, $P < 0.01$), 99.9% (***, $P < 0.001$) and 99.99% (****, $P < 0.0001$) confidence intervals.

Line 152: You performed three PCR assays solely for species identification. Please summarize the results of these assays and include them in lines 152-153.

Answer: I have added the following information in this section:

3.1 Identification Results of Isolated Strains

The identification results showed that all 34 clinical isolates were *A. fumigatus*. All the sequences of the analyzed strains were uploaded to GenBank (PP069948-PP070390).

Line 155: Consider including a table that shows the MIC distribution for each antifungal agent without the combination treatments. For example, include the number of isolates with an MIC of 0.25 for posaconazole, etc.

Answer: Thank you for your suggestion. I have added the following table:

Table 1. Minimum inhibitory concentration (MIC) values distribution of tested strains

Azole	MIC	Non-knockout Strains	Knockout Strains	Total
ITC	0.125	2	4	6
	0.25	17	12	29
	0.5	6	13	19
	1	7	1	8
	2	3	1	4
	4	3	0	3
VOR	0.125	0	2	2
	0.25	12	4	16
	0.5	21	11	32
	1	5	12	17
	2	0	1	1
	4	0	1	1
POS	0.125	2	6	8
	0.25	17	10	27
	0.5	8	13	21
	1	7	2	9
	2	4	0	4

Line 156-161: Specify the number of isolates for each group, e.g., clinical isolates (n=36), defective strains (n=31), MFS transport strains (n=20), etc. This will make the data easier to understand.

Answer: Thank you for your suggestion. I have added the information as follows: BBM alone did not show anti- *A. fumigatus* activity. Specifically, in clinical isolates (n=37), 0.125 to 4 µg/mL for ITC, 0.25 to 1 µg/mL for VRC, and 0.125 to 2 µg/mL for POS; Among the defective strains in MFS transporter (n=20), 0.125 to 0.5 µg/mL for ITC, 0.125 to 1 µg/mL for VRC, and 0.125 to 1 µg/mL for POS. Among the defective strains in ABC transporter (n=5), 0.25 to 0.5 µg/mL for ITC, 0.125 to 1 µg/mL for VRC, and 0.125 to 0.5 µg/mL for POS. Among TAC cycle-related gene deficient strains (n=6), 0.125 to 2 µg/mL for ITC, 0.25 to 4 µg/mL for VRC, and 0.25 to 1 µg/mL for POS (Table 1). When BBM was used in combination with ITR, VOR, POS, it synergised mainly with POS: high rates of synergistic effects were shown WT, 35 clinical isolates (94.4%), and 30 genetically defective strains (96.8%) (total synergistic rate 95.7%).

Line 178: You are reporting individual isolate data here; please also include the results for the three groups tested.

Answer: Thank you for your suggestion. I have added the results for the three groups tested.

Fig.1 Changes in *cdr1B* expression after drug stimulation

Note: ^a, Measurements taken after the combined stimulation with POS; ^b, Measurements taken after the combined stimulation with POS and BBM.

(****, $P < 0.0001$.)

Line 179: Explain why *cdr1B* is likely a target for the BBM-posaconazole combination rather than BBM-itraconazole or BBM-voriconazole.

Answer: Since BBM primarily exhibited synergy when combined with POS, and the deletion of *cdr1B* resulted in the loss of this synergy, it suggests that *cdr1B* plays a crucial role in the BBM-POS combination. In contrast, BBM combined with ITC or VOR did not show significant synergy, and the deletion of *cdr1B* did not lead to either synergy or antagonism. Furthermore, although POS, ITC, and VOR all belong to the azole class of antifungals, their structural and pharmacological properties differ. I have revised this section as follows:

As BBM did not exhibit significant synergy with ITC or VOR, but showed strong synergy with POS, and since the synergy observed in the BBM-POS combination was lost in the $\Delta cdr1B$, *cdr1B* is likely a key target in the synergistic mechanism of BBM and POS. Include, from the 36 clinical isolates, Af05 and Af08 did not produce a synergistic effect, possibly due to inherently low

expression levels in these strains.

Line 204: Again, summarize the results for the three groups first. This will provide context for your focus on the BBM-posaconazole combination over the other drug combinations.

Answer: I have added a summary of the results at the beginning of this section:

Only the BBM-POS combination demonstrated a significant synergistic effect. Therefore, we further investigated the impact of the BBM-POS combination on ROS levels.

Line 285-290 and 292-295: These sections repeat the same content. Please select one version to include in the manuscript as the conclusion.

Answer: I have removed the redundant content from the discussion section and retained the conclusion section to avoid repetition.

Line 298: Justify why ethical approval was not obtained, considering that clinical isolates were used in the study.

Answer: Thank you for your feedback. I have added the relevant ethical statement:

Ethics approval: This study was performed in line with the principles of the Declaration of Helsinki. Approval was granted by the Ethics Committee of Jingzhou Hospital Affiliated to Yangtze University.

Minor comments:

Line 30: Where do the 66 strains and defective strains come from? Are they also clinical strains, or were they isolated from the environment?

Answer: The 66 strains include wild-type (WT), clinical isolates, and genetically defective strains. The clinical isolates were obtained from Jingzhou Central Hospital. Af293 was purchased from the Fungal Genetics Stock Center (FGSC). The genetically defective strains were constructed based on the A1160 (Δ KU80, *pyrG*) tool strain, which was also obtained from FGSC. I have supplemented this information in the supplementary materials under "**2. Strain Identification**":

The cultured fungal specimens were taken and preliminarily identified according to the morphologic characteristics. Fungal DNA was extracted by MolPure Fungal DNA Kit and further

amplified the ribosomal DNA transcriptional spacer internal transcribed spacer (ITS), beta-tubulin and calmodulin genes (Table S2)[2-5], PCR was performed using the following parameters: 3 min at 95°C, followed by 35 steps of 1 min at 95°C, 1 min at 58.5 °C and 1 min at 72 °C, and then a final 10 min at 72 °C. The final products were sequenced by Biocompany [BioEngineering (Shanghai) Co., Ltd], and finally, the sequence was blasted in NCBI GenBank. The definitive identification of the *Aspergillus* isolates was accomplished by comparing the sequences with relevant reference sequences in GenBank using the nucleotide BLAST system (<https://blast.ncbi.nlm.nih.gov/Blast.cgi>).

Table S2. Primer sets and corresponding amplification targets

Target gene	Primer	Primer DNA sequence(5'-3')
ITS	ITS1	TCCGTAGGTGAACCTGCGG
	ITS4	TCCTCCGCTTATTGATATGC
calmodulin	cmd5	CCGAGTACAAGGAGGCCTTC
	cmd6	CCGATAGAGGTCATAACGTGG
beta-tubulin	Bt2a	GGTAACCAAATCGGTGCTGCTTTC
	Bt2b	ACCCTCAGTGTAGTGACCCTTGCC

Line 32: When you refer to "these resistant strains," which specific strains are you talking about?

Answer: Thank you for your clarification. These strains refer to the non-synergistic strains, specifically Af05 and Af08. I have revised this section as follows:

Results: BBM alone showed no significant antifungal activity. BBM combined with POS exhibited synergy against 66 strains (95.7%), while two clinical isolates (Af05/Af08) and one defective strain ($\Delta cdr1B$) showed no synergy. Synergy with ITC was observed in 3 strains (4.3%), and none with VOR. In the non-synergistic Af05 and Af08 strains, the expression of the *cdr1B* gene was significantly lower compared to wild-type (WT) strains. ROS levels increased significantly in WT with POS and BBM combination therapy, but not in the defective strains. Glucose uptake was also reduced in the POS-BBM combination.

Line 32-33: Under the methods section, please specify the number and types of the 69 *A. fumigatus* strains tested, including whether they are wild-type strains.

Answer: Thank you for your suggestion. I have made the necessary additions as follows: A total of 69 strains of *A. fumigatus* were tested, including the wild-type strain WT, 37 clinical isolates, and 31 genetically deficient strains (Table S1). The genetically deficient strains can be divided into those missing MFS transporters (20), ABC transporters (5), and genes related to the TCA cycle (6). *Candida parapsilosis* (ATCC 22019) and *Aspergillus flavus* (ATCC 204304) were included to ensure quality control. The construction of the deficient strains was achieved using high-throughput gene knockout based on the principle of homologous recombination[29]. Details on the identification of clinical isolates and the construction of genetically deficient strains are provided in the supplementary file.

Line 63: Consider moving Figure 1 to the supplementary section.

Answer: Thank you for your suggestion. I have moved Figure 1 to the supplementary file.

Line 73: Clarify whether berbamine hydrochloride (BBM) was effective when used alone in treating cancer, or only when combined with other agents.

Answer: Thank you for your suggestion. I have added the necessary clarification: Recently, a number of studies have demonstrated the anti-tumour effects of BBM in a number of different cancers, including myeloma, prostate cancer, lung cancer, pancreatic cancer, liver cancer and gastric cancer[19-21]. However, there have been no reports on the combination of BBM with other drugs. Furthermore, studies have demonstrated that BBM can effectively induce intracellular reactive oxygen species (ROS) levels[22-26].

Line 83: What is a standard strain, and how does it differ from the others?

Answer: Thank you for your correction. *A. fumigatus* AF293 is a clinical isolate, and it is a widely recognized standard reference strain in *A. fumigatus* research. This strain was fully sequenced in 2005 (Nature, 2005), and its genomic data includes 8 chromosomes with approximately 29.4 Mb of DNA, encoding over 9,900 predicted genes.

Line 84: Please explain what constitutes a standard strain. Aren't the wild-type and genetically deficient strains also isolated from patients? If these are environmental strains, please mention this

as well.

Answer: A standard strain refers to a microbial strain that has been certified by authoritative organizations, possesses a well-defined genetic background, exhibits a stable phenotype, and serves as a benchmark for scientific research, experimental controls, or quality assurance. These strains are typically preserved and distributed by international microbial resource centers such as ATCC, DSMZ, and CGMCC to ensure data consistency and reproducibility across laboratories worldwide. For example, ATCC 22019 and ATCC 204304 are considered standard strains. Among the strains used in this study, Af293 is a clinical isolate. I have revised this section as follows:

A total of 69 strains of *A. fumigatus* were tested, including the wild-type strain WT, 37 clinical isolates, and 31 genetically deficient strains (Table S1). The genetically deficient strains can be divided into those missing MFS transporters (20), ABC transporters (5), and genes related to the TCA cycle (6). *Candida parapsilosis* (ATCC 22019) and *Aspergillus flavus* (ATCC 204304) were included to ensure quality control. The construction of the deficient strains was achieved using high-throughput gene knockout based on the principle of homologous recombination[29]. Details on the identification of clinical isolates and the construction of genetically deficient strains are provided in the supplementary file.

Line 90-91: Rephrase to clarify that sequencing of the beta-tubulin and calmodulin genes was performed to confirm species identity. Also, provide detailed procedures in the supplementary section.

Answer: Thank you for your suggestion. We extracted DNA from the strains and sequenced the amplified regions of ITS, beta-tubulin, and calmodulin genes to confirm the species identity. The sequencing results were compared in the NCBI database, and the ITS sequencing results have been uploaded to the database. I have provided detailed procedures in the supplementary file:

2. Strain Identification

The cultured fungal specimens were taken and preliminarily identified according to the morphologic characteristics. Fungal DNA was extracted by MolPure Fungal DNA Kit and further amplified the ribosomal DNA transcriptional spacer internal transcribed spacer (ITS), beta-tubulin and calmodulin genes (Table S2)[2-5], PCR was performed using the following parameters: 3 min

at 95°C, followed by 35 steps of 1 min at 95°C, 1 min at 58.5 °C and 1 min at 72 °C, and then a final 10 min at 72 °C. The final products were sequenced by Biocompany [BioEngineering (Shanghai) Co., Ltd], and finally, the sequence was blasted in NCBI GenBank. The definitive identification of the *Aspergillus* isolates was accomplished by comparing the sequences with relevant reference sequences in GenBank using the nucleotide BLAST system (<https://blast.ncbi.nlm.nih.gov/Blast.cgi>).

Table S2. Primer sets and corresponding amplification targets

Target gene	Primer	Primer DNA sequence(5'-3')
ITS	ITS1	TCCGTAGGTGAACCTGCGG
	ITS4	TCCTCCGCTTATTGATATGC
calmodulin	cmd 5	CCGAGTACAAGGAGGCCTTC
	cmd 6	CCGATAGAGGTCATAACGTGG
beta-tubulin	Bt2a	GGTAACCAAATCGGTGCTGCTTTC
	Bt2b	ACCCTCAGTGTAGTGACCCTTGGC

Line 93: Please write out the full name of VOR.

Answer: Thank you for your correction. I have now written out the full name of VOR as voriconazole in the manuscript.

Line 94: Include the company and country of purchase for the drugs.

Answer: Thank you for your suggestion. I have added the company and country information as follows: All tested agents including BBM, itraconazole (ITC), VOR (voriconazole), posaconazole (POS) were purchased in powder form from Aladdin Biochemical Technology Co., Ltd., Shanghai, China, and diluted in dimethyl sulfoxide($\geq 99.9\%$) as stock solutions (6400 $\mu\text{g}/\text{mL}$).

Line 96: Is it SDA or SAB? What does the "B" in SAB stand for? Please ensure this is corrected throughout the manuscript.

Answer: Thank you for your suggestion. It is SAB, which stands for Sabouraud Dextrose Broth. The "B" in SAB refers to "Broth", indicating its liquid form. In this study, solid medium was prepared by adding 2% agar to SAB. I have ensured that this terminology is correctly used

throughout the manuscript.

Line 101: Specify where "As described"

Answer: Thank you for your correction. I have revised this section as follows: The working concentration ranges of BBM, ITR, VRC and POS were 0.5–32 µg/mL, 0.06–8 µg/mL, 0.06–8 µg/mL and 0.03–4 µg/mL, respectively. A 50 µL of BBM with serial dilutions were inoculated in horizontal direction and another 50 µL of azoles with serial dilutions were inoculated in vertical direction on the 96-well plate, which contained 100 µL prepared inoculum suspension. Interpretation of results was performed after incubation at 35°C for 48h. The MICs applied for the evaluation of effects against *A. fumigatus* was determined as the lowest concentration resulting in 100% inhibition of growth[32].The FICI as calculated by the formula: $FICI=(Ac/Aa)+(Bc/Ba)$, where Ac and Bc are the MICs of antifungal drugs in combination, and Aa and Ba are the MICs of antifungal drugs A and B alone[33]. A FICI of ≤ 0.5 is classified as synergy, a FICI of >0.5 to ≤ 4 indicates no interaction (indifference), and a FICI of >4 indicates antagonism[34]. All tests were performed in triplicate.

Line 115: What did the *A. fumigatus* control group receive during the experiment?

Answer: Thank you for your feedback. I have revised this section comprehensively as follows:

2.4. Detection of ROS content

A. fumigatus conidia were collected after 48 hours of incubation on SAB solid medium at 35°C and resuspended in 1640 liquid medium to a final concentration of 5×10^4 cfu/mL. All samples were incubated at 37°C in a shaking incubator at 130 rpm for 16 hours. During the incubation period, drug treatments were applied as needed. At the end of the incubation, 2,7-dichlorodihydrofluorescein diacetate (DCFH-DA, Yeasen Biotechnology, Shanghai, China) was added to each sample and incubated at 37°C for 1 hour to detect intracellular ROS levels. Flow cytometry data acquisition was performed using a Beckman Cytomics FC 500 BD FACSCanto II, and data analysis was conducted using FlowJo v10 software. The excitation

wavelength was set at 488 nm, and the emission wavelength at 525 nm.

Line 114-115: Rephrase "culturing at 37°C." Do you mean that the isolates were cultured and incubated at 37°C?

Answer: Thank you for your feedback. I have revised this section comprehensively as follows:**2.4.**

Detection of ROS content

Line 127: Suggest you remove "refer to 2.4 for grouping" and instead add "for each group" at the end of the sentence in line 129.

Answer: Thank you for your suggestion. I have made the following revision: Total RNA was extracted using TRIeasy (Yeasen Biotechnology, Shanghai, China) and reverse-transcribed into cDNA using a Hifair® III 1st Strand cDNA Synthesis SuperMix (Yeasen Biotechnology, Shanghai, China) for each group as per the manufacturer's recommendation.

Line 127: Change to TRIeasy.

Answer: Thank you for your suggestion. I have made the following revision: Total RNA was extracted using TRIeasy (Yeasen Biotechnology, Shanghai, China) and reverse-transcribed into cDNA using a Hifair® III 1st Strand cDNA Synthesis SuperMix (Yeasen Biotechnology, Shanghai, China) for each group as per the manufacturer's recommendation.

128: Add "as per the manufacturer's recommendation."

Answer: Thank you for your suggestion. I have made the following revision: Total RNA was extracted using TRIeasy (Yeasen Biotechnology, Shanghai, China) and reverse-transcribed into cDNA using a Hifair® III 1st Strand cDNA Synthesis SuperMix (Yeasen Biotechnology, Shanghai, China) for each group as per the manufacturer's recommendation.

Line 129: Provide more details about the method in the supplementary section.

Answer: Thank you for your suggestion. I have added the detailed information in the supplementary section as follows:

3. Gene expression analysis

A. fumigatus conidia were collected after 48 hours of incubation on SAB solid medium at 35°C and resuspended in 1640 liquid medium to a final concentration of 5×10^4 cfu/mL. All samples were incubated at 37°C in a shaking incubator at 130 rpm for 16 hours. During the incubation period, drug treatments were applied as needed. Total RNA was extracted using TRIeasy (Yeasen Biotechnology, Shanghai, China) and reverse-transcribed into cDNA using a Hifair® III 1st Strand cDNA Synthesis SuperMix (Yeasen Biotechnology, Shanghai, China). Concentrations and quality were also determined using Nanodrop one (ThermoFisher Scientific). PCR was performed using the following parameters: 5 min at 25°C, 15 min at 55°C and 5 min at 85°C. RT-qPCR was performed in triplicate on Real-Time Quantitative Thermal Cycler (MA-6000, Yarui Biotechnology, China) with Hieff qPCR SYBR Green Master Mix (Yeasen Biotechnology, Shanghai, China). The relative gene expression level ($2^{-\Delta\Delta CT}$) was calculated using the actin housekeeping gene control[6]. All primers used for gene expression analysis are listed in Supplemental Table S3.

Line 132: What are the "mutual control groups" referred to here?

Answer: I have rephrased this part to clarify the statement: Total RNA was extracted using TRIeasy (Yeasen Biotechnology, Shanghai, China) and reverse-transcribed into cDNA using a Hifair® III 1st Strand cDNA Synthesis SuperMix (Yeasen Biotechnology, Shanghai, China) for each group as per the manufacturer's recommendation. Continuing to detect the expression of *cdr1B* gene under different treatments (Table S3). The relative gene expression level ($2^{-\Delta\Delta CT}$) was calculated using the actin housekeeping gene control[35].

Line 134: Correct "fumigates" to fumigatus.

Answer: Thank you for your correction. I have updated the text to use fumigatus instead of fumigates.

Line 135: Suggest you remove "refer to 2.4 for grouping" and instead add "for each group" at the end of the sentence.

Answer: Thank you for your suggestion. I have made the revision here.

Line 135: Again, correct "fumigates" to fumigatus.

Answer: Thank you for your correction. I have updated the text to use fumigatus instead of fumigates.

Line 146: The microplate reader was read at the excitation wavelength of...

Answer: Thank you for your suggestion. I have made the correction as follows: The microplate reader was read at the excitation wavelength of 529 nm.

Line 180: Include, from the 36 clinical isolates, Af05...

Answer: Thank you for your suggestion. I have revised this section as follows: Include, from the 36 clinical isolates, Af05 and Af08 did not produce a synergistic effect, possibly due to inherently low expression levels in these strains.

Line 183: After Af293, include "(standard strain)" for clarification.

Answer: Thank you for your suggestion. I have made the revision as follows: Therefore, the expression levels of the *cdr1B* in the clinical isolates Af05 and Af08 were also examined. The results showed that compared to WT and isolated strain Af293, the initial expression levels of the *cdr1B* in Af05 and Af08 were significantly reduced.

Line 192-202: I would suggest moving this information to the supplementary section.

Answer: Thank you for your suggestion. I have moved this section to the supplementary section.

Line 255: Why have you separated these isolates into pathogenic and knockout categories? Does this mean that since these strains have been knocked out, they are no longer pathogenic? Are the genes in question responsible for *A. fumigatus* virulence or antimicrobial resistance?

Answer: These strains are primarily divided into clinical isolates and genetically deficient strains. The knockout strains were constructed using the KU80 strain as the parent, and specific genes were knocked out to create the deficient strains. Detailed methods are provided in the supplementary materials. The genetically deficient strains still retain their pathogenicity. When

these knockout strains were subjected to combination antifungal susceptibility testing, the synergistic effect disappeared after the relevant genes were knocked out, suggesting that these genes may be potential targets for the synergistic effect.

Sincerely,

Yi Sun

Department of Dermatology, Jingzhou Hospital Affiliated to Yangtze University, Hubei Provincial
Clinical Research Center for Diagnosis and Therapeutics of Pathogenic Fungal Infection, Jingzhou,
Hubei Province, 434100, China

Tel:+86 18071883358

E-mail:jzxyysy@163.com

Re: Spectrum03184-24R1 (*In vitro* Interactions of Berbamine Hydrochloride and Azoles against *Aspergillus fumigatus*)

Dear Dr. Yi Sun:

Your manuscript has been accepted, and I am forwarding it to the ASM production staff for publication. Your paper will first be checked to make sure all elements meet the technical requirements. ASM staff will contact you if anything needs to be revised before copyediting and production can begin. Otherwise, you will be notified when your proofs are ready to be viewed.

Sincerely,
Vera Tesic
Editor
Microbiology Spectrum

Reviewer #1 (Comments for the Author):

The suggestions have been addressed by the authors.